# Dominant resistance and negative epistasis can limit the co-selection of de novo resistance mutations and antibiotic resistance genes

Andreas Porse[1,2], Leonie J. Jahn[1,2], Mostafa M.H. Ellabaan[1] & Morten O.A. Sommer [1✉]

To tackle the global antibiotic resistance crisis, antibiotic resistance acquired either vertically by chromosomal mutations or horizontally through antibiotic resistance genes (ARGs) have been studied. Yet, little is known about the interactions between the two, which may impact the evolution of antibiotic resistance. Here, we develop a multiplexed barcoded approach to assess the fitness of 144 mutant-ARG combinations in *Escherichia coli* subjected to eight different antibiotics at 11 different concentrations. While most interactions are neutral, we identify significant interactions for 12% of the mutant-ARG combinations. The ability of most ARGs to confer high-level resistance at a low fitness cost shields the selective dynamics of mutants at low drug concentrations. Therefore, high-fitness mutants are often selected regardless of their resistance level. Finally, we identify strong negative epistasis between two unrelated resistance mechanisms: the *tetA* tetracycline resistance gene and loss-of-function *nuo* mutations involved in aminoglycoside tolerance. Our study highlights important constraints that may allow better prediction and control of antibiotic resistance evolution.

---

[1] Novo Nordisk Foundation Center for Biosustainability, Technical University of Denmark, Kgs. Lyngby, DK-2800, Denmark. [2] These authors contributed equally: Andreas Porse, Leonie J. Jahn. ✉email: msom@bio.dtu.dk

Bacterial evolution is driven by two main mechanisms: selection of genomic mutations (vertical evolution) and the acquisition of foreign DNA through horizontal gene transfer (HGT). These two modes of evolution provide the genetic plasticity that allows bacteria to inhabit virtually all ecological niches on the planet and underlines the emergence of antibiotic-resistant pathogenic bacteria increasingly compromising our ability to treat infections[1,2].

The acquisition and retention of genes or mutations that confer antibiotic resistance is strongly affected by the genetic context. For example, chromosomal mutations leading to antibiotic resistance most often occur at multiple loci[3], and interactions between different mutations can increase resistance levels or reduce the fitness cost of initially costly mutations[4,5]. Similarly, the transfer of most plasmids, including those carrying individually cloned ARGs, initially has a negative impact on host fitness[6–10], and these costs may be ameliorated by compensatory mutations in either the plasmid or the chromosomes[5,6,11,12]. Intriguingly, while ARGs have evolved to function in a broad range of genomic contexts, and genomic mutations leading to resistance are host-specific, the resulting resistance mechanisms often overlap[13]. For example, a protein targeted by an antibiotic may be altered via mutation(s) to avoid inhibition; or by horizontal acquisition of effectors that modify or replace the protein target, thereby rendering the cell resistant[14]. Taken together, these observations suggest that chromosomal mutations and HGT events could interact to contribute to the development of antibiotic resistance.

In addition to promoting antibiotic resistance development in circulating strains, epistatic interactions between resistance determinants may also lead to increased resistance or collateral sensitivity during sequential treatment regimes in individual patients[15–18]. For example, mutations associated with aminoglycoside resistance confer sensitivity towards multiple antibiotic drug classes, including beta-lactams, quinolones, and tetracyclines[17–19], and this sensitivity is thought to arise from a reduction in the proton motive force (PMF) that reduces the uptake of aminoglycoside, while simultaneously decreasing multidrug efflux[18].

Thus, it is reasonable to hypothesize that horizontally and vertically acquired resistance factors could interact in a similar way to affect antibiotic resistance or sensitivity. However, little is known about the direct interactions that may occur between ARGs and mutations that confer antibiotic resistance. Whereas no previous work has looked into the interactions between ARGs and resistance mutations directly, a study by Silva et al.[20] investigated the fitness effects of combining conjugative plasmids with resistance-conferring mutations in the *gyrA*, *rpoB*, and *rpsL* genes of *Escherichia coli*[20]. This study suggested that the coexistence of these mutations and conjugative plasmids had an overall positive effect on bacterial fitness. However, it is unclear to what extent the observed epistatic effects are caused by the plasmid-encoded ARGs or other components of the large plasmid backbones and whether these interactions are maintained during antibiotic selection.

Given the importance of chromosomal mutations and acquired resistance genes in the evolution of multidrug resistance, there is a need to assess both modes of resistance simultaneously on a broader scale[21]. To shed light on the specific epistatic interactions between resistance genes and resistance mutations, we develop a multiplexed competition approach to assess the fitness of a panel of ARG-mutant combinations against a representative set of antibiotics at a variety of clinically relevant concentrations.

## Results
### ARG-mutant interactions promote differential mutant selection. An important aspect of resistance evolution is how different genotypes interact during antibiotic treatment. Such epistatic interactions may cause preferential selection or counterselection of certain genotypic combinations in a manner that would not be predicted from their individual effects. To study epistatic interactions between resistance mutations and resistance genes, we selected 11 different chromosomally barcoded mutants, adapted to resist different antibiotics, and 11 synthetic ARGs[10,22]. The mutants and ARGs were chosen to represent a wide range of mutational targets and biochemical resistance mechanisms (Supplementary Tables 1, 2). To assess the effect of each mutational background on the function of each ARG, we created a combined library containing each barcoded host transformed with each ARG, comprising a total of 144 combinations of ARGs and resistance mutations, including the background strain (wild-type, WT) and an empty vector control. To assess the effect of potential interactions on resistance phenotypes, we pooled all lineages transformed with the same ARG and subjected each ARG-mutant pool to selection by antibiotics representing eight different clinically important drug classes, including both bactericidal and bacteriostatic antibiotics, at 11 different concentrations (Fig. 1).

Initially, all ARG-mutant pools were subjected to antibiotics at concentrations just above the minimal inhibitory concentration (MIC) of the WT strain carrying the empty vector backbone, which was generally close to the clinical breakpoints for the selected drugs (Supplementary Table 6), and the highest drug concentration with growth were selected for further characterization for each antibiotic tested. These concentrations were chosen to monitor all possible interactions while minimizing the chance of extinction of mutants due the low level of resistance generally conferred by mutations, or in case of further reductions caused by strong negative epistasis. The barcoded genomic region of each ARG-mutant pool was sequenced after 24 h of incubation in each condition to quantify the relative abundance of each ARG-mutant combination; reflecting the fitness of each ARG and resistance mutation combination under each selective condition (Fig. 2). A control without antibiotics was included to quantify the fitness of each combination in the absence of drug selection (Supplementary Fig. 1).

In the absence of antibiotic selection we observe no changes in the mutant abundance profile with the introduction of ARGs (ANOVA, $p > 0.05$), highlighting that the resistance genes confer no or only minor fitness costs to the bacteria. With antibiotic selection we observe 10 of 88 cases in which the mutant abundance profile is significantly (ANOVA, $p < 0.05$, Bonferroni corrected) changed by the introduction of a particular ARG. It should be noted, that the majority of ARG-mutant combinations did not affect each other even under selecting condition. Under these conditions, mutants with the highest fitness in presence of the antibiotic and a MIC above the drug exposure were dominating. In most cases, those mutants were adapted to the antibiotic they were exposed to. However, in a few instances we also observed the selection of a mutant that was adapted to another drug but shows cross resistance to the drug tested. For example in Cefepime mutants evolved to Doxycycline were preferentially selected even though the Cefepime evolved lineage had a higher MIC in Cefepime and comparable fitness in LB. This could be explained by changes in fitness in the presence of the antibiotic or by population dynamics between different mutants in the mutant pool. Unexpected selection and cross resistance between unrelated conditions has been previously reported for Cefepime, highlighting previous adaptations and population dynamics can impact selection patterns[22].

From the ten observed interactions, the most common change (eight out of ten cases) was the selection of a mutant distribution similar to the nonselective (LB) condition. This could result from

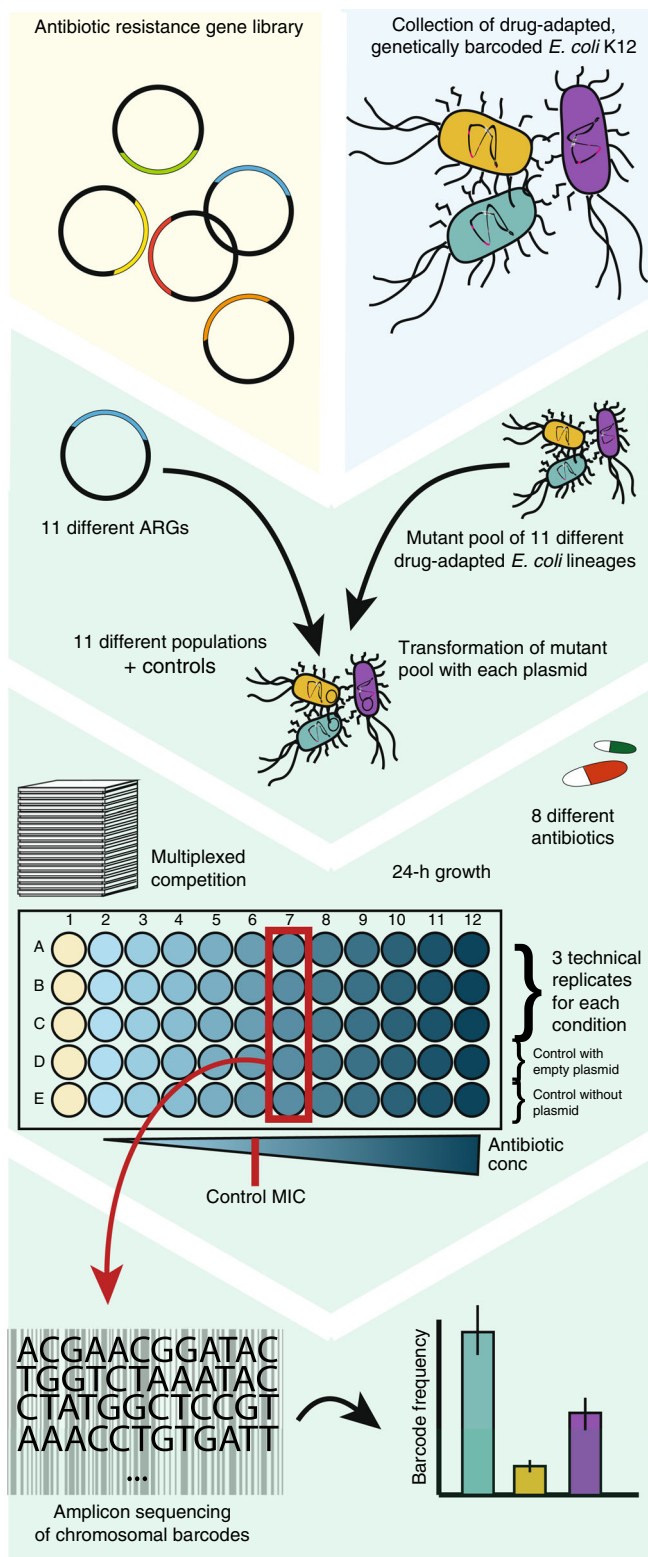

**Fig. 1 Experimental overview.** Eleven pools of 11 different mutants were individually transformed with 11 ARGs to create 121 unique ARG-mutant combinations. In addition, 23 combinations containing either the empty vector or the background *E. coli* MG1655 WT strain were made. The mutant pools containing each ARG were subjected to eight different drugs, as well as an antibiotic-free condition. At least two replicates of the individual pools were then sequenced to determine the abundance of each mutant barcode.

the ARG shielding the effect of the antibiotic on the mutant pool when it confers resistance to the given antibiotic. One case of mutant dominance was also observed for the *qnrS1* gene present in the *gyrA* ciprofloxacin resistant background. In this case the antibiotic resistance gene did not shield the selection of the mutant. Finally, a case of strong negative epistasis was observed for the tetA gene in the amikacin-evolved background (Fig. 2).

In summary, these results suggest that while many chromosomal resistance mutations and ARGs did not significantly affect each other, strong and diverse interactions do occur.

**ARGs tend to dominate mutants within their resistance range.** The initial results suggested that the selection of mutants in the presence of a given ARG would depend largely on the ARGs ability to confer resistance to the presented drug challenge (Fig. 2). The selection patterns observed suggests that ARGs dominate the resistance phenotype of the mutants and that, accordingly, the selection of mutants would depend on their fitness in the condition without antibiotics (Supplementary Fig. 2).

To investigate this phenomenon further, we determined mutant selection patterns in the presence or absence of the ARGs $bla_{TEM-219}$, *floR*, and *qnrS1* at concentrations ranging from sub-MIC levels to 256 times the WT MIC of cefepime, chloramphenicol, and ciprofloxacin, respectively. When no antibiotic was present, the fittest mutants (LB, CIP, and TMP evolved) were selected, as highlighted also in Fig. 2. From the selection patterns observed at the different drug concentrations, it is clear that mutants were selected for at very low (sub-MIC) drug concentrations in the absence of a protective ARG (Fig. 3). A pattern of concentration-dependent mutant selection was observed, which corresponded well with the relative fitness cost of each mutant in the absence of antibiotics, as well as the respective antibiotic resistance profile (Supplementary Fig. 2 and Supplementary Table 3). Mostly, the mutants were selected at a respective concentration where they had the highest fitness in LB combined with a MIC above the drug exposure level. For example, above the MIC of chloramphenicol the mutants evolved to Doxycycline are dominating (Fig. 3b) as observed previously (Fig. 2).

For mutants carrying the $bla_{TEM-219}$ or *floR* resistance genes, the presence of the ARG substantially enhanced the resistance of all of the mutants of the pool, resulting in the selection of mutants primarily based on their fitness rather than on their resistance level to the drug for the lower drug concentrations included here. This is consistent with the ARGs shielding, or dominating, the impact of the antibiotic within a specific concentration range corresponding to the resistance range of the ARG (Supplementary Fig. 3A, B); after which additive effects of ARGs and mutants are observed. At drug concentrations just above the MIC, still the fittest mutants (LB, CIP, and TMP evolved) were selected as also noted in Fig. 2. However, at high drug concentrations, when the antibiotic exposure approached the resistance level conferred by the ARG, the resistance gene no longer shielded the mutants, and differential selection of mutants resistant to the specific antibiotic was observed (Fig. 3a, b), suggesting an additive effect of mutants and ARGs at these drug concentrations. Here, the mutant selection resembled the selection at lower drug concentrations without the ARGs present.

Interestingly, the *qnrS1* ciprofloxacin resistance gene confers lower resistance to ciprofloxacin than the low-cost *gyrA* (ciprofloxacin-adapted) mutation and does not provide substantial protective benefits to sensitive hosts (Fig. 3c). Because the *gyrA* mutant is highly resistant and its fitness cost is low, the resistance gene adds only a minimal amount of resistance to the mutant pool. Therefore, only a marginal shift in mutant

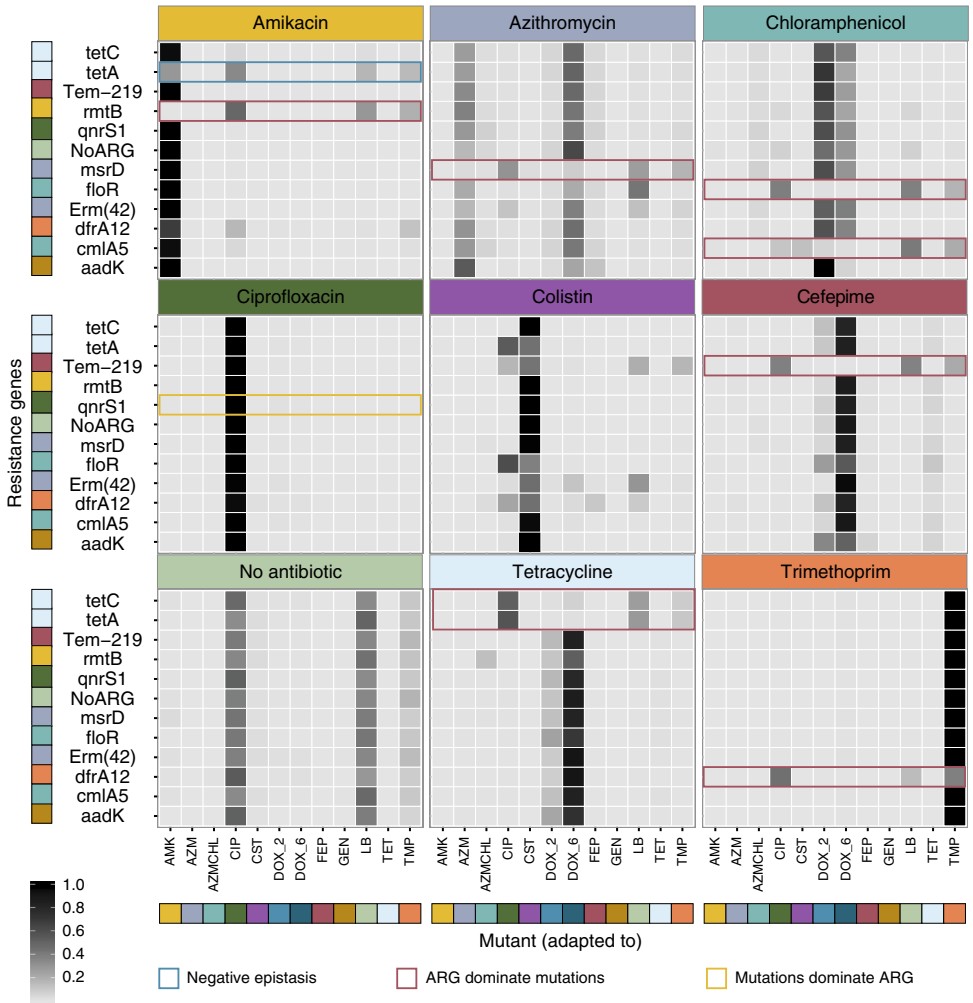

**Fig. 2 Selective patterns of mutant pools transformed with antibiotic resistance genes.** Each ARG-mutant pool was subjected to different drugs (indicated above the panels) at a concentration twice that of the WT MIC. The shading of the heatmap tiles illustrates the relative abundance of each mutant within each ARG-associated mutant pool. Highlighted combinations (except yellow) show the significant deviations (ANOVA, $p < 0.05$) from the null-hypothesis that the ARG has no influence on the mutant distribution across at least two replicates. Combinations highlighted in red illustrate the cases where the ARG shields mutants that would otherwise be selected. Blue highlights situations in which an ARG that does not confer resistance to the antibiotic tested reduces the selection of the mutant; suggesting a negative interaction between the mutant and the ARG. Cases highlighted in yellow demonstrate the selection of the resistant mutant despite the presence of a resistance gene conferring resistance to the antibiotic tested, suggesting a dominant effect of the mutant over the resistance gene. ARG and mutant labels are colored according to the expected main resistance conferred. AMK (amikacin), AZM (azithromycin), AZMCHL (azithromycin and chloramphenicol), CIP (ciprofloxacin), CST (colistin), DOX_2 (doxycycline—low resistance), DOX_6 (doxycycline—high resistance), FEP (cefepime), GEN (gentamycin), LB (Medium without antibiotics), TET (tetracycline), and TMP (trimethoprim). Source data are provided as a Source Data file.

selection is observed in the presence of *qnrS1*, resulting in a small additive effect dominated by the resistant mutant (Supplementary Fig. 3). Explaining, why no difference in selection was noted in Fig. 2 for the qnrS1 gene.

To validate selected experiments performed using our pooled competition approach, we additionally performed traditional pairwise competitions as previously described (Supplementary Fig. 4 and Supplementary Table 4)[20]. From the results, we observe similar dynamics to those obtained in our pooled competitions, e.g., strong negative epistasis of *tetA* in the amikacin adapted mutant competed in amikacin. Similarly, we could confirm the strong negative epistasis observed between mutants and ARGs for the antibiotic conditions used in Fig. 2. We also confirmed the concentration-dependent selection of the *dox6*, *azmchl* and *fep* mutants observed in Fig. 3a, where the *dox6* mutant wins the *dox–fep* pairwise competition only at lower cefepime concentrations and the azmchl mutant is slightly fitter than both dox6 and fep

mutants at higher (4 µg/ml) cefepime concentrations (Supplementary Fig. 4B).

In addition, the pairwise competitions agree with our pooled fitness data where the ARG-mutant combinations did not show significant ($p > 0.05$, stats test) epistatic interactions in the absence of antibiotics (Supplementary Fig. 4A and Supplementary Table 4).

Taken together, these results suggest that mutant fitness and the resistance capacity of an ARG are the main factors determining which mode of resistance is selected at a particular concentration and that these parameters dictate when additive effects occur. The fact that most ARGs confer high levels of resistance means that resistance mutations may often be redundant when ARGs are present.

**Negative epistasis of TetA and mutations in the *nuo* genes.** Interestingly, we discovered an example of strong negative

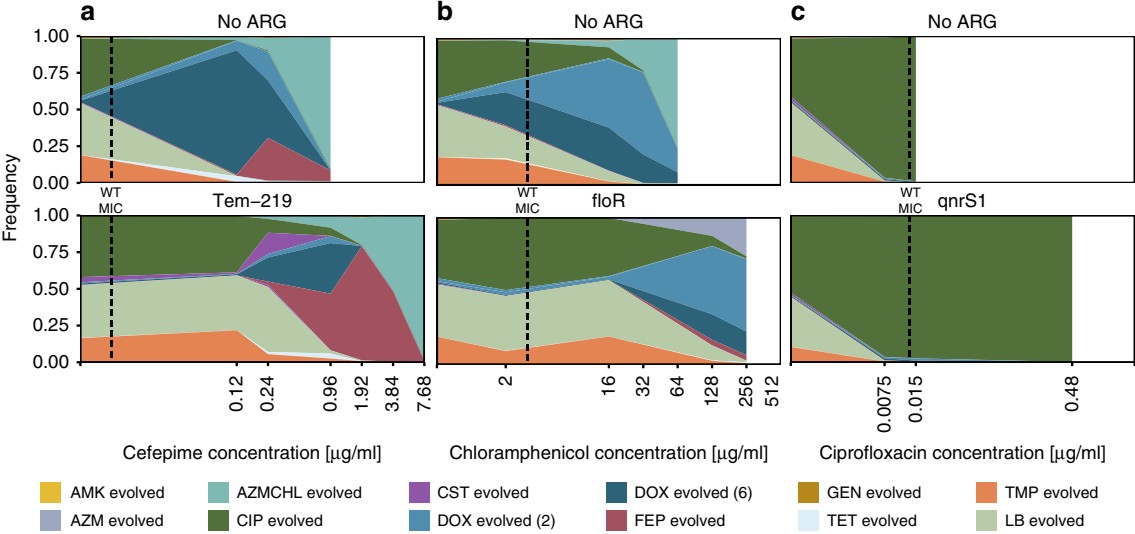

**Fig. 3 Concentration-dependent mutant selection with and without ARG shielding.** Mutant pools with and without expression of Tem-219 beta-lactamase (**a**) or the FloR efflux pump (**b**) at different concentrations of either cefepime (Tem-219) or chloramphenicol (FloR) were sequenced to determine the mutant selection patterns. Panel **c** Shows the dynamics of mutant selection in the presence of the *qnrS1* conferring low resistance compared with the *gyrA* mutation. Mutants were evolved in AMK (amikacin), AZM (azithromycin), AZMCHL (azithromycin and chloramphenicol), CIP (ciprofloxacin), CST (colistin), DOX_2 (doxycycline—low resistance), DOX_6 (doxycycline—high resistance), FEP (cefepime), GEN (gentamycin), LB (Medium without antibiotics), TET (tetracycline), and TMP (trimethoprim). Source data are provided as a Source Data file.

epistasis between the tetracycline efflux pump TetA and the amikacin-evolved mutant. The amikacin-evolved mutant was not selected to the same extend in amikacin (2X WT MIC) when the TetA gene was present (>75% decrease in relative abundance compared with the control) as when the remaining ARGs were present demonstrating a negative epistatic interaction (Fig. 2).

To identify the causal mutation of the TetA-induced aminoglycoside sensitivity, we used multiplexed automated genome engineering (MAGE) to simultaneously reverse all individual mutations of the amikacin-resistant mutant to their respective WT sequence in the presence of the *tetA* gene. The Amk3 mutant had point mutations in *nuoH*, *cpxR*, *crr*, *fusA*, and *rffG*, as well as a small deletion in *lrhA*. While the effect of *crr*, *rffG*, and *lrhA* mutations are not clear, the *fusA*, *cpxR*, and *nuoH* mutations are often associated with resistance towards aminoglycosides[18,19,23]. The *nuo* genes encode subunits of NADH:quinone oxidoreductase I, which maintains the PMF to fuel respiration[24]. The PMF is required for aminoglycoside uptake, and mutations of the *nuo* genes can lead to a decreased PMF, resulting in a high tolerance towards aminoglycosides[18]. After one MAGE cycle, a *tetA*-carrying MAGE pool containing the reversed mutants was selected on amikacin to identify the causal mutation leading to the TetA-induced amikacin sensitivity. Sequencing the targeted loci of four surviving colonies revealed that the *nuoH* mutation had been reversed to the WT sequence, suggesting that this mutation interacts antagonistically with *tetA* in the presence of amikacin (Supplementary Table 5).

To test if this negative interaction with TetA might apply to the other genes in the *nuo* cluster, we selected a different aminoglycoside-resistant mutant (Amk4), which carried mutations in *nuoF* and *fusA* only. This mutant showed similar properties to the Amk3 (*nuoH*) mutant, including reduced growth at sub-MIC concentrations of aminoglycosides, which was especially pronounced for streptomycin, when *tetA* was present (Fig. 4a and Supplementary Fig. 5). This demonstrates that mutations in several genes in the *nuo* pathway have negative epistatic interactions with *tetA*. We further tested the involvement of the nuo mutant in this sensitive phenotype, by conducting a pairwise competitive fitness experiment, where the tetA-carrying

nuoF and fusA mutant was competed against a tetA-carrying mutant with the same fusA background but with the nuoF allele reverted to WT (Fig. 4b). Here, the *nuoF* mutant showed a marked competitive disadvantage at streptomycin concentrations above 4 µg/ml, with complete suppression at 32 µg/ml. In addition, the presence of tetracycline did not have a material effect on the selection patterns.

We speculated that the effect of *tetA* on the *nuo* mutants was connected to the PMF, in which case it would be dependent on the external proton concentration (pH). To test this idea, we measured the effect of the growth medium pH on the growth of the *tetA*-carrying *nuoF* mutant at different streptomycin concentrations (Fig. 4c). By measuring the growth reduction relative to growth in the *tetA*-free background at three streptomycin concentrations, we observed a pH dependent increase in the *tetA*-conferred sensitivity.

To investigate the broader implications of such negative epistasis on the occurrence of the *nuo* resistance mutations and *tetA* in natural isolates, we analyzed their co-occurrences in all 13,500 sequenced *E. coli* genomes. In these genomes, *tetA* was never observed together with the specific nucleotide polymorphisms in *nuoF* and *nuoH* that were observed in this study, nor with any other mutation expected to result in loss of function in the *nuo* pathway (e.g., stop codons and INDELS). For example, *tetA* was observed 377 times and *nuoF* mutants 661 times; however, they were never observed together, although they would be expected to co-occur in 18 genomes by chance.

Taken together the experimental and bioinformatic results suggest that the presence of *tetA* alters the resistance level of aminoglycoside resistance mutants, carrying mutations in the nuo operon, in a PMF-dependent manner and that such interaction may limit their co-selection in natural *E. coli* isolates."

**Several transporters induce streptomycin sensitivity under high expression.** Given the strong effect of *tetA* in a genetic context in which a *nuo*-family gene is mutated when exposed to streptomycin, we hypothesized that such negative epistatic interactions with the *nuo* genes could be a more general attribute

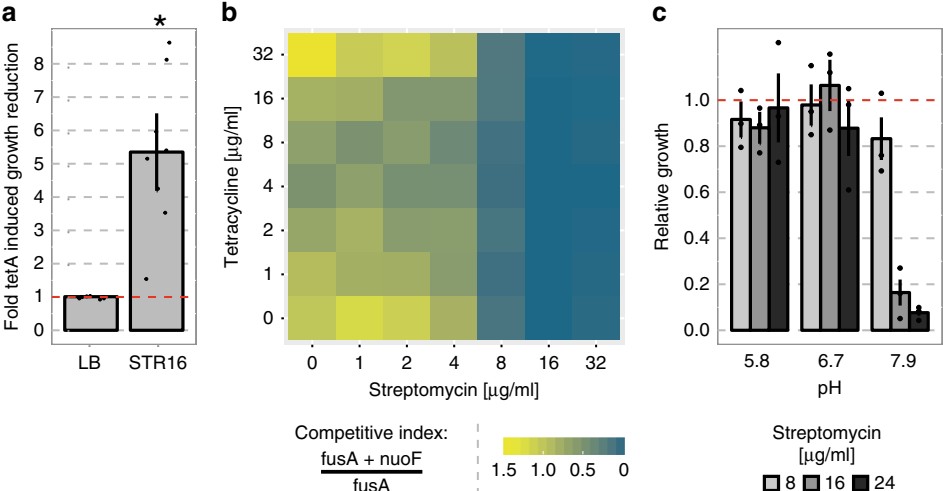

**Fig. 4 The presence of TetA induces *nuoF*-dependent sensitivity to aminoglycosides. a** Fold *tetA*-induced growth reduction of the Amk4 mutant grown in media in the presence (STR16) or absence (LB) of a sub-MIC concentration of streptomycin (16 μg/ml). Error bars display the standard error of the mean ($n = 8$). The red dashed line shows the baseline growth without antibiotics. The asterisk indicates statistical significance compared with the baseline (Wilcoxon rank sum test, $p < 0.05$). **b** Fitness landscape of *nuoF*-reversed mutants competing against their ancestor at different concentrations of tetracycline and streptomycin. A more blue color, as opposed to yellow, indicates outcompetition of the *nuoF* mutant. **c** Relative growth of the *tetA*-carrying *nuoF* mutant compared with the corresponding mutant carrying an empty vector at different streptomycin concentrations and pH values. Error bars show the standard error of the mean ($n = 3$) and the red dashed line indicates the value of the WT to which the remaining values are normalized. Source data are provided as a Source Data file.

of antibiotic efflux pumps. To test this, we expressed the *tetC* and *tetG* tetracycline pumps and the chloramphenicol efflux pumps *cmlA5* and *floR*, which share 39%, 43%, 17%, and 18% amino acid sequence identity with *tetA*, respectively, in the *nuoF* background. Although these efflux pumps vary substantially in their primary structure, they all belong to the major facilitator superfamily (MFS) of transporters that comprises the largest group of acquired antibiotic transporters[25]. These were expressed in the *nuoF* background, and the sensitivity of these strains was compared with that of the empty vector control when grown in media containing streptomycin. For the *cmlA5*, *floR*, and *tetC*, we observed significant sensitivity towards streptomycin (Wilcoxon rank sum test, $p < 0.05$), demonstrating that this phenotype is not exclusive to *tetA* (Fig. 5a). These data suggest that MFS transporters other than TetA may facilitate the uptake of compounds in an unspecific manner.

Therefore, to assess the effect of the selected efflux pumps on general membrane permeability, we measured the accumulation of Hoechst 33342 in the *nuoF* mutant expressing the same MFS transporters as tested in the streptomycin sensitivity assay. Compared with the empty vector control, all efflux pumps increased the membrane permeability of the indicator dye Hoechst 33342 by 17.8% on average (Wilcoxon rank sum test, $p < 0.001$) (Fig. 5b). This increase in membrane permeability, induced by a diverse set of efflux pumps, suggests that the uptake is not determined by their specific functionality but rather by a more general effect, e.g., their perforation of the inner membrane, which may supplement the intrinsic routes of aminoglycoside uptake.

## Discussion

Both genomic mutations and HGT contribute to the emergence of multidrug-resistant bacteria, yet the relative contribution of each evolutionary mechanism to clinical resistance problems remains unclear. Using a multiplexed phenotyping approach, we were able to simultaneously test multiple ARG-mutant combinations under multiple conditions to elucidate epistatic interactions that may direct antibiotic resistance evolution.

A previous study by Silva et al.[20] analyzed fitness effects for combinations of conjugative plasmids with *gyrA*, *rpoB*, and *rpsL* mutants and found strong epistatic effects. Our study focused specifically on ARG-mutant interactions and did not identify strong epistatic interactions. A possible explanation for this discrepancy is that the observed epistatic effects do not stem from ARG-mutant interactions, but rather from the remaining portion of the large plasmid backbones. This notion is supported by previous studies of plasmid-host evolution that describe interactions between the host chromosome and several plasmid components not related to antibiotic resistance[6,26,27]. Contrary to the study by Silva et al.[20] we specifically assessed the ARG-mutant interaction and did not find any significant effect on fitness in the absence of antibiotics. While it is possible that our 24 h competition assay could not detect minor effects on combined ARG-mutant fitness, we were able to detect the subtle fitness differences between low-cost mutants, e.g., in those with mutations in *folA* and *gyrA*, compared with the susceptible ancestor (Supplementary Fig. 2), suggesting that that potential effects missed by our assay where minor.

The coexistence of ARGs and chromosomal mutations that confer resistance towards the same condition are more likely to be selected for if they interact in a favorable manner to increase overall resistance or fitness. In such cases, we observed an additive effect of ARG-mutant combinations only when antibiotic concentrations where high (generally above the clinical breakpoint). The negative epistatic interaction between ARGs and mutants, at lower drug concentrations observed here, may therefore encourage reversion to susceptible genotypes when ARGs are present in environments exposed to drug concentrations within their resistance capacity, where most mutational resistance phenotypes do not add to survival.

Notably, we also observed a case of strong negative epistasis between two unrelated resistance mechanisms (*nuoF* or *nuoH* mutants with TetA). The counterselection of *nuoF* and *nuoH* mutants in the presence of *tetA* and aminoglycoside selection is an interesting example of antagonism between genes with unrelated resistance functionalities. TetA has previously been shown

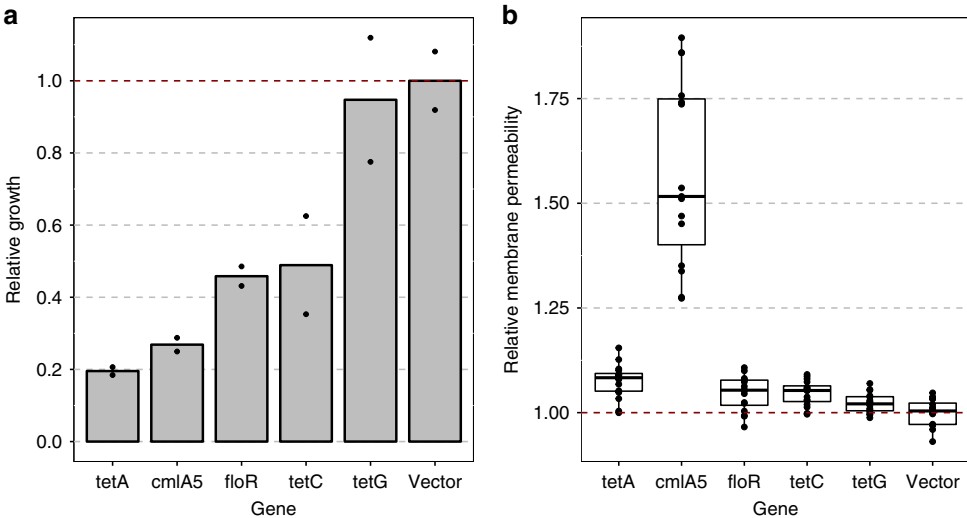

**Fig. 5 Effect of different MFS efflux pumps on streptomycin sensitivity and membrane permeability in the Amk4 mutant.** Various genes encoding MFS pumps were independently expressed in the *E. coli nuoF + fusA* mutant strain. Growth (**a**) and membrane permeability (**b**) were measured and are displayed relative to those of the same *E. coli* strain carrying an empty vector. Membrane permeability was measured as the increase in fluorescence resulting from Hoechst 33342 uptake. The bars show the mean of two replicates (**a**), and the boxplot the interquartile range between the first and third quartiles and median (internal line) with dots representing individual fluorescence measurements ($n = 16$) (**b**). The red dashed lines indicate the values of the emty expression vector to which the remaining values are normalized. Source data are provided as a Source Data file.

to increase the uptake of different compounds, including aminoglycosides, when highly expressed[28,29]. However, the increase in the sensitivity of aminoglycoside-resistant mutants has not been described before. Importantly, loss-of-function *nuo* mutants are known to arise during aminoglycoside treatment and have shown a strong increased tolerance towards aminoglycosides in addition to a more general drug persistence phenotype that is likely due to the decrease in respiration resulting from a decreased PMF[23,30]. Because the presence of MFS efflux pumps such as TetA may lower the PMF threshold needed for aminoglycoside tolerance, a broad range of efflux pumps could render selection of *nuo* mutations less favorable during drug treatments (Fig. 5a). This antagonistic relationship was supported by our genomic mining of *E. coli* genomes, where *tetA* and loss-of-function *nuo* mutants were not observed together. As drug tolerance is believed to be a critical first step towards drug resistance[31,32], incorporating knowledge of how ARG-mutant interactions may curb this transition will be necessary to fully understand resistance evolution.

It has previously been shown that high *tetA* expression increases aminoglycoside uptake without altering the membrane potential[28]. This, along with the data presented here, indicates that the increased sensitivity towards aminoglycosides is not due to *tetA* altering the membrane potential itself, but rather to the cells expressing *tetA* increasing aminoglycoside uptake with the PMF gradient to a higher extent than cells lacking *tetA*. While we observed a high sensitivity towards especially streptomycin, the degree of sensitivity likely relates to properties of both the drug and efflux pump, as is evident from the diverse sensitivity profiles of the tested efflux pumps and drugs.

In conclusion, while the coexistence of most interaction pairs was not constrained, our data demonstrates existence of several interactions between ARGs and chromosomal resistance mutations that may affect their dissemination. Further studies should seek to understand antibiotic resistance in a broader genomic context as well as more clinically relevant settings. Yet, we believe that studying the interactions between ARGs and resistance mutations will further improve our ability to understand, predict, and prevent antibiotic resistance evolution.

## Methods

**Strains and plasmids.** Twelve isolated colonies of chromosomally barcoded *E. coli* MG1655 K12 (DSMZ, DSM 18039) lineages were chosen from the endpoint of a large-scale adaptive evolution experiment based on the diversity of their mutation profile[22]. The mutations evolved as a response to the exposure of the strains to increasing drug concentrations. Eleven strains were adapted to different antibiotics and carried between one and six chromosomal mutations (Supplementary Table 1) affecting, among other processes and parameters: drug uptake, efflux, the drug target, and global transcription levels. One strain was adapted in LB medium as a control to include effects of adaptations to the growth medium. The 12 strains were pooled in equal cell numbers, and aliquots of this mutant pool were transformed via electroporation with a medium copy number plasmid (p15A replicon) carrying one of 11 different ARGs (Supplementary Table 1)[10]. The 11 ARGs were chosen to confer resistance to a broad selection of antibiotics via different mechanisms, and were expressed from a weak promoter to simulate realistic ARG expression levels (Supplementary Table 2)[10]. Two aliquots of the mutant pool served as controls: one was transformed with the empty vector, while the other remained untransformed. A selection of the included ARGs encoding efflux pumps was also cloned into a high-copy vector (pZE21)[33] and transformed into the Amk4 (*fusA + nuoF*) mutant strain.

**Multiplexed competition experiment.** First, 150 μl of medium with or without added antibiotics was inoculated with ~$10^6$ cells from an overnight culture of the transformed mutant pools and incubated for 18 h at 37 °C without shaking. Each mutant pool was subjected to a twofold concentration gradient ranging over 11 different concentrations of 8 different antibiotics, resulting in more than 3000 competition experiments in at least two replicates. This approach was used to mimic the methodology by which the individual MIC values were measured. The antibiotics used cover a wide range of clinically important drug classes and mechanisms, including both bactericidal and bacteriostatic antibiotics (Supplementary Table 6). In addition, cells were grown in LB alone. For all conditions, 40 μg/ml zeocin was added to prevent plasmid loss, and no bias in mutant selection was observed at this concentration of zeocin (Supplementary Fig. 1).

**Pairwise competition experiments.** Head to head competition experiments were performed in accordance with the methodology of Silva et al.[20]. Strains were grown O/N in LB and diluted $10^{-4}$ before mixing in 10 ml LB. All competitions were done in three biological replicates. After initial plating to quantify starting numbers, the competitions were incubated at 37 °C for 24 h with shaking. The reference strains used here was *E. coli* K12 MG1655ΔlacZ (created by introducing a nonsense mutation in the *lacZ* gene) and CFU of the competing strain and reference was enumerated as blue and white colonies, respectively, on LB plates containing X-Gal (Sigma). The *lacZ* mutant did not show a significant fitness cost ($P = 0.57$) when competed against its ancestor. Four competitions were conducted with direct competition of mutants *dox6* against *fep* and *fep* against *azmchl* in cefepime and these were distinguished based on differences in their resistance profiles.

**Amplicon sequencing**. Amplicon sequencing of the mutant pools was performed before and after each transformation. In addition, at least two replicates of each mutant pool were sequenced after growth at antibiotic concentrations twofold the WT MIC, as well as at the highest antibiotic concentration that displayed growth. Based on interesting trends derived from the sequencing data, the DNA of samples incubated at additional concentrations was also sequenced for certain ARG-mutant pools. The pools of sequenced samples and conditions under which they were incubated are listed in Supplementary Table 5. Amplicon sequencing of the barcoded region in the genome was initiated by a PCR using the following primers: Fwd_Primer: TCGTCGGCAGCGTCAGAGTGTATAAGAGACACAATGACCGG GCTTTCCGC and Rev_Primer: GTCTCGTGGGCTCGGAGATGTGTATAAGA GACAGGGATGCTATGGTTTCAGG, which contain the adaptors amplified by the NEBNext Multiplex Oligos for Illumina sequencing (New England BioLabs, MA, USA). These PCR products were subsequently used for indexing PCRs. Samples were purified and pooled into aliquots with equal DNA concentrations, and then sequenced on an Illumina MiSeq. The barcode frequencies for each condition were determined as the number of reads containing each barcode relative to the total number of barcoded reads.

**Mutant repair using MAGE**. The mutations found in the amikacin-evolved (Amk3) mutant were repaired using single-stranded oligo recombineering[34]. The AMK mutant was transformed with the pMA7 vector, which carries the λ recombineering system[35]. The recombineering system was induced in midlog phase by adding 0.2% w/v arabinose and incubating for an additional 15 min. Competent cells were prepared by three rounds of washing at 4 °C, and electroporation of a pool of oligos was performed (Supplementary Table 5). The transformed cultures were allowed to recover for 4 h before plating on 10 μg/ml amikacin LB plates to select resistant clones.

**Growth rate determination**. Optical density (OD) measurements were conducted in 96-well plates containing 150 μl of LB medium per well using the *ELx808* plate reader (BioTek, USA). The OD at 600 nm ($OD_{600}$) was measured over 5 min intervals for a maximum of 20 h, and the plates were incubated at the medium shaking setting at 37 °C between measurements. Media with varying pH values were prepared by diluting HEPES buffer to a concentration of 50 mM in LB and subsequently adjusting the pH with NaOH and HCl. The final pH was measured just before inoculation. Sensitivity measurements of MFS efflux pumps in the *fusA* + *nuoF* background were performed at different concentrations of streptomycin (0–64 μg/ml) in twofold increments, and the exponential growth rate was normalized to that of the antibiotic-free (LB) samples for each concentration. The average sensitivity was then calculated as the average relative reduction in the LB-normalized growth rate relative to the growth rate of the empty vector control at all concentrations where the empty vector control displayed growth (<32 μg/ml).

**2D competitive fitness assay**. The *tetA*-carrying *fusA* + *nuoF* mutant and the *nuoF*-repaired mutants were transformed with *gfp*- and *rfp*-expressing pZE21 plasmids, respectively[33]. Each strain was grown to an $OD_{600}$ of 0.5, diluted 1000× and mixed equally before inoculation into a 96-well clear-bottomed black plate containing a 2D gradient of twofold dilutions of streptomycin and tetracycline in 200 μl LB medium per well. Fluorescence (at 528 and 615 nm with excitation at 485 and 580 nm for GFP and RFP, respectively) and $OD_{600}$ were measured for 10 h using the *Synergy H1* plate reader (BioTek, USA). The RFP to GFP signal ratio after 8 h of growth is reported as the competitive index.

**Membrane permeability assay**. The DNA-intercalating dye Hoechst 33342 changes its fluorescent properties upon DNA binding and has been applied to measure the membrane permeability of bacterial cells[18,36,37]. The accumulation of Hoechst 33342 is sensitive to the activity of endogenous multidrug efflux systems, such as the AcrAB–TolC system of *E. coli*, and to membrane-piercing structures, such as porins, that ease dye entry[36,37]. Cultures were grown to an $OD_{600}$ of 0.3, and 100 μl of each strain was transferred into a clear-bottomed black plate. Using a 96-channel pipette (INTEGRA VIAFLO 96), 100 μl of additional LB medium containing 5 μM Hoechst 33342 was simultaneously added to each well. The plate was incubated at 37 °C with shaking in a *Synergy H1* plate reader (BioTek, USA), and fluorescence and $OD_{600}$ were read every 5 min. Fluorescence was read from the bottom of the plate using 355 and 450 nm excitation and emission filters, respectively[36]. The membrane permeability was then calculated as the area under the fluorescence/OD curve.

**Resistance gene and mutation co-occurrence analysis in sequenced *E. coli* genomes**. Approximately 13,500 complete *E. coli* genomes were downloaded from the NCBI RefSeq database (June 2018). BLAST analysis was performed to identify ARGs (clustered at 95% identity) and mutations associated with the genes mutated in this study (Supplementary Table 1). BLAST identification of *tetA* and the *nuo* genes involved in mutational resistance was performed at 99% ID. INDELs and stop codons within a reading frame were regarded as having similar (loss-of-function) outcomes. The statistical analysis was performed using the *Cooccur* package of R to conduct the co-occurrence analysis[36]. Here, the *p* value cutoff for

interaction-type classification was Bonferroni-adjusted based on the number of total interactions.

**Reporting summary**. Further information on research design is available in the Nature Research Reporting Summary linked to this article.

## Data availability
The authors declare that all the relevant data are provided in this published article and its Supplementary Information files, or are available from the corresponding author on request. The source data underlying Figs. 2–5 and Supplementary Figs. 1–5 are provided as a Source Data file.

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

## Acknowledgements

M.O.A.S. acknowledges support from the EU H2020 ERC-20104-STG LimitMDR (638902) and the Danish Council for Independent Research Sapere Aude programme DFF -4004-00213. The Lundbeck Foundation under grant agreement R140-2013-13496 and the Novo Nordisk Foundation under NFF grant number NNF10CC1016517.

## Author contributions

A.P., L.J.J., and M.O.A.S. designed the study. A.P. and L.J.J. performed the experiments and analyzed the data with input from M.O.A.S. M.M.H.E. constructed the genome database and performed B.L.A.S.T. searches. A.P. and L.J.J. wrote the paper with input from M.O.A.S.

## Competing interests

The authors declare no competing interests.
