## [Peer Review File · Nature Communications]

Reviewers' comments:

Reviewer #1 (Remarks to the Author):

I think the idea behind this study is sound, interesting, and should be pursued. As clearly explained in the manuscript, a few years ago, Silva et al (ref. 20) showed that the coexistence of resistance-conferring chromosomal mutations and conjugative plasmids (carrying (antibiotic-resistance-genes, ARGs, often acquired by HGT) had an overall positive effect on bacterial fitness (positive epistasis). Now, it would be interesting to know what kind of interactions there are between ARGs alone (i.e., without the rest of the conjugative plasmid) and chromosomal mutations. The scientific problem is important and interesting. However, I have serious concerns about the way the manuscript is written. In several places, I could not understand what has been done experimentally or how to interpret results.

Chromosomal barcoding as a tool for multiplexed phenotypic characterization of laboratory evolved lineages, developed by the same group last year, is now applied to study interactions between chromosomal mutations and ARG. However, the exact procedure for Fig 2, for example, is not clearly explained. Take, for example, the first line, left, in Fig2, when Amikacin was used and the case of the ARG tetC. Did the authors mix all barcoded mutants, each one also harboring the tetC gene? Another problem, also in Fig.2, is that one has to take too much time to interpret the figure according to the meaning of each gene on the Y-axis (and keep using "Supplementary Table 2. Antibiotic resistance genes."). There should be a way to clarify this information in the figure. The discussion about Fig.2 is extremely concise.

In Fig.3, I think that there is a mistake: the FloR and Tem-219 cases should be exchanged, I guess (that is (Fig3A) and (Fig3B) are exchanged). But again, it is quite difficult to understand the meaning of this Fig.. This figure is almost not discussed. For example, when the antibiotic concentration is zero, one should expect that the barcoded mutants left should be the ones with higher fitness as shown in "Supplementary Figure 2. Unselected mutant fitness", right? This should be explained. In Fig. 3B (should be FloR, I guess, as explained above), when the concentration of chloramphenicol is 2 ug/ml is this what we can see in Fig2? In that case, are is green zone the CIP-evolved and the grey zone the LB-evolved, and is this the reason why these squares are darker in Fig.2? All these things should be clearly explained.

Assuming that my interpretation of Fig. 2 is correct and that Fig3A and Fig3B are exchanged, I now have some scientific concerns.

- 1) In Figure 2, why did you choose the concentration 2xMic? Why this value? What did you observe with other concentrations.
- 2) In Figure 2, why didn't the authors discount what happened with the LB situation?
- 3) In some places, MIC is referred and, in others, ug/ml is mentioned.
- 4) In Fig 4A, I think the legend is not very clear, given what it is in the Y-axis. In the legend, one can read "Doubling-time of the Amk4...", but then, the Y-axis of Fig 4A is "Fold tetA induced growth reduction". Is it "doubling time" or the factor by which the doubling time increased?
- 5) In Fig 4C, one can see the relative growth reduction of the tetA-carrying nuoF mutant. But is this really what the authors meant? In the figure, we see that at high pH, bars are lower. Does this mean that reduction is lower (that is, increase in the tetA-conferred sensitivity with increasing pH)?
- 6) I find it interesting that there is an overall positive effect when mutations and conjugative plasmids are together (Silva et al, ref 20) but that now the authors observe negative epistasis between mutations and ARGs. Does this imply that the backbones of plasmids are, somehow, helping ARGs to be maintained in bacterial cells (conferring lower fitness cost)?
- 7) Title: I think that the title of this manuscript is misleading. Given that, when an ARG arrives into a cell, it is often travelling in a conjugative plasmid. In this case, there is no limitation of co-selection of vertically and horizontally acquired antibiotic resistance factors: ARGs do not arrive alone.

Reviewer #2 (Remarks to the Author):

The manuscript studies the interaction between chromosomal mutations and horizontally acquired resistant determinants in the evolution of antibiotic resistance. The authors combined 11 different chromosomally barcoded *E. coli* mutants adapted to different antibiotics with 11 different antibiotic resistance genes (ARGs) present on a medium copy plasmid and measured the fitness of the

mutant-ARG combinations in the presence of 8 different antibiotics. At relatively low antibiotic concentrations (twice the MIC of the wild type carrying the empty vector) they identified significant interactions for approx. 12% (10 out of 88) of the mutant-ARG combinations and in 80% of these cases

ARGs dominated the resistance phenotype of the mutant, shielding the selective effect of the antibiotic on the mutant pool. The authors reach the conclusion that at low antibiotic concentrations the resistance determinant with the highest resistance level will dominate and shield the selection dynamics of the other resistant determinant, while at high antibiotic concentrations both determinants will be selected for. In one case they identified negative epistasis between two unrelated resistance mechanisms, the *tetA* efflux pump and loss of function *nuo* mutations involved in aminoglycoside resistance. By the mining of 13500 *E. coli* genomes they found that *tetA* was never observed with any loss of function mutations in the *nuo* pathway, supporting the negative epistasis.

Overall, I appreciate the systematic approach that integrates vertically and horizontally acquired resistance determinants to study resistance evolution and I find the results to be of potential broad interest.

However, the clinical relevance of the work is questionable due to the low antibiotic dosage employed. Also, I wish to highlight that a paper with similar goals have recently been published (Silva et al, *Plos Genetics*, 2011). Given major differences in the methods used to evaluate epistasis between the two studies, it is difficult to reconcile why different conclusions were reached.

Major comments:

Pooled competition assays are appropriate as a screening method to have a first and rude approximation of bacterial fitness. However, for the identification of exact epistatic interactions, pairwise

competition assays (i.e. wild-type versus antibiotic resistant bacteria) are needed to reach precise relative fitness values both in the presence and absence of antibiotic. Based on these precise fitness values, the nature of interactions (negative, positive or sign epistasis) can be determined and deviation from the null model (additive interaction) should be evaluated statistically. I find it critical to verify the nature of epistatic interactions in a rigorous manner, see e.g. Silva et al, *Plos Genetics*, 2011. I believe this should be done for several mutant-plasmid pairs.

The conclusions drawn from a competition assay performed at twice the wild type MIC antibiotic concentration. Is this relevant? I think it is important to repeat the high-throughput pooled competition assay closer at much higher dosages. Does epistasis depend on the dosage used? If not, this should be justified in systematic manner. I understood correctly, mutant pools were sequenced at the highest antibiotic concentration too, so this can be easily verified. In addition, pairwise epistatic interactions should be determined as well at these higher antibiotic concentrations to reach clinical relevance with the results. Related to clinical relevance, I missed the information whether the resistance level of the used antibiotic adapted lines is clinically relevant (above the clinical breakpoint) or not. Same question applies to the ARGs.

The authors claim in the legend of figure 2 that "the ARG shields the selection of the mutant that would otherwise be selected." It is not clear what the initial expectation about the selection of mutants is, if there is any expectation. Do we expect those mutants to be selected i) that were already adapted to the given antibiotic (this happens in 5 out of the 8 cases), ii) that have the highest fitness in the absence of any antibiotic or iii) that show cross-resistance to the given antibiotic and have a high fitness (e.g. doxycycline mutants)? For example, what can be the possible explanation that cefepime adapted mutants are not selected in the presence of cefepime but doxycycline mutants are, when cefepime mutants have the highest cefepime MIC and the fitness of the two mutants is very similar in the absence of antibiotic selection. Similar question applies to tetracycline adapted lines. To have a clear picture, I would suggest the de novo construction of nuoH and nuoF single mutants starting from the wild type background and verify in the conventional way that indeed they show negative epistasis with tetA. Then test the sensitivity of these mutants against different aminoglycosides in the presence of TetA to clarify if the observed negative epistatic interaction applies generally to aminoglycosides or not.

When the authors tested the effect of the growth medium pH on the tetA-conferred streptomycin sensitivity, they claim that tetA-conferred sensitivity increased with increasing pH. However, based on figure 4C, with the exception of the highest streptomycin concentration, no clear trend can be observed. I would suggest repeating the experiment using a broader pH range. It also remains an open question why these particular (5.8, 6.7, 7.9) pH values were chosen and it is not clear what data is visualized on the y axis of the figure. What does the red dotted line show? The growth of the mutant carrying an empty vector? If yes, then the relative growth of tetA-carrying nuoF mutant is shown on the y axis.

Minor comments:

- 1) How exactly was the relative abundance of each mutant within each ARG-associated mutant pool calculated?
- 2) It is not clear what NoVec stands for in supplementary figure 1, if relative abundances of ARG-mutant combinations are shown normalized to the fitness of mutants carrying the empty vector.
- 3) In figure 3 Tem-219 and floR are interchanged. It would be informative to highlight the 2xMIC concentration in this figure.
- 4) Supplementary figure 4 is very difficult to interpret. Please provide more information in the figure legend, for example what does the red dotted line represent and what exactly is shown on the y axis?
- 5) What was the reason to incubate the cells without shaking in the competition experiment?

Reviewer #3 (Remarks to the Author):

In general, the idea of looking for genetic interactions between resistance genes and chromosomal resistance mutations is quite interesting. The authors performed a screen for this based on previously generated experimental evolution, and then focused on a specific interaction between tetA resistance gene and nuo mutations involved in aminoglycoside tolerance. The computational analysis showing that these two mutations never co-occur despite expectation was quite

interesting.

My biggest issue with this paper is that the experiments in Figure 1 suggest that interactions between vertically and horizontally are not very common, but the title and abstract seem to imply the opposite. I am concerned the result of this is to oversell the generality of the results. The specific interaction explored, however, is quite interesting, and the paper would likely be far better off claiming that well-supported finding as a main result and not attempting to overgeneralize.

Specific Comments:

- Fig 2 was hard to understand, it took a few attempts to really get it. Also, showing log (fold change in frequency) for each strain might be easier to interpret than relative abundance. It might also be useful to have a metric for comparing abundance profiles.
- Line 133: I'm afraid I do not know what it means to "dominate in an additive fashion" -- it's clear what it means when the effect of one mutation dominates another, which is what I believe the authors intend here. There is a small bit of explanation on lines 159-60, but I think it would help a broader audience understand the results if the authors took some care to be a bit more explicit and precise in their claims here.
- In Fig 3A, 3B: authors claim that the mutant pool profile below a certain amount of antibiotic does not depend on antibiotic concentration. While the claim is plausible, the authors interpolate between data points that are quite widely spaced (between 0 and 2, 2 and 16 in 3A, 0 and 12 in 3B). It would be useful to have data for the intermediate points, or to weaken the claim.
- Line 201: Sentence starting with "To confirm the involvement" was a confusingly worded and it took some time to understand what strains were used in the competition experiment.
- Line 205: "Additionally, the presence of tetracycline did not have a material effect on the selection patterns": there appears to be no data to support this claim.
- 2D competitive fitness assay (Figure 4B and 397-404): While the results are strong, to establish the effect beyond a doubt it would be good to repeat the experiment with dye-swapped strains. I do not think this is strictly necessary, however, as the likelihood of there being a differential effect on drug resistance between *gfp* and *rfp* is minimal.
- Lines 230-232 ("These results suggest... limit their co-selection in natural *E. coli* isolates"): That the lack of co-occurrence is due to aminoglycoside selection is a bit of a leap and is not supported. As many things depend on PMF, this pattern could be due to any number of selective pressures.
- Line 343: Why were these cultures grown without shaking? This is pretty nonstandard, as it allows within-well differentiation due to nutrient (and oxygen) gradients and wall-bound growth. While I don't expect this to bias the results too much, it may make them harder to reproduce.

Answers to Reviewer comments:

Reviewer #1 (Remarks to the Author):

I think the idea behind this study is sound, interesting, and should be pursued. As clearly explained in the manuscript, a few years ago, Silva et al (ref. 20) showed that the coexistence of resistance-conferring chromosomal mutations and conjugative plasmids (carrying (antibiotic-resistance-genes, ARGs, often acquired by HGT) had an overall positive effect on bacterial fitness (positive epistasis). Now, it would be interesting to know what kind of interactions there are between ARGs alone (i.e., without the rest of the conjugative plasmid) and chromosomal mutations. The scientific problem is important and interesting.

We thank the reviewer for her/his interest in our study.

However, I have serious concerns about the way the manuscript is written. In several places, I could not understand what has been done experimentally or how to interpret results.

We are grateful for the reviewer's comments on our manuscript and hope that the implementation of changes makes our experimental setup and conclusions clearer to our readers.

Chromosomal barcoding as a tool for multiplexed phenotypic characterization of laboratory evolved lineages, developed by the same group last year, is now applied to study interactions between chromosomal mutations and ARG. However, the exact procedure for Fig 2, for example, is not clearly explained. Take, for example, the first line, left, in Fig2, when Amikacin was used and the case of the ARG tetC. Did the authors mix all barcoded mutants, each one also harboring the tetC gene?

We thank the reviewer for highlighting insufficient explanations for Figure 2. It is correct that we mixed all barcoded mutants each harbouring the resistance gene. In order to clarify this, we added the following explanation to the text from line 85-86:

*"To assess the effect of each mutational background on the function of each ARG, we created a combined library containing each barcoded host transformed with each ARG, comprising a total of 144 combinations of ARGs and resistance mutations, including the background strain (wild-type, WT) and an empty vector control. To assess the effect of potential interactions on resistance phenotypes, we **pooled all lineages transformed with the same ARG and subjected each ARG-mutant pool** to selection by antibiotics representing eight different clinically important drug classes, including both bactericidal and bacteriostatic antibiotics, at 11 different concentrations (Figure 1)."*

Another problem, also in Fig.2, is that one has to take too much time to interpret the figure according to the meaning of each gene on the Y-axis (and keep using "Supplementary Table 2. Antibiotic resistance genes."). There should be a way to clarify this information in the figure. The discussion about Fig.2 is extremely concise.

In order to make Figure 2 easier to interpret, we implemented a color scheme to visualize the connection between mutants, ARGs and antibiotics more clearly. This color scheme has also been applied to Figure 3.

In addition we added to the figure legend and main text to better describe the figure.

Updated figure legend:

Figure 2. Selective patterns of mutant pools transformed with antibiotic resistance genes. Each ARG-mutant pool was subjected to different drugs (indicated above the panels) at a concentration twice that of the WT MIC. The shading of the heat-map tiles illustrates the relative abundance of each mutant within each ARG-associated mutant pool. Highlighted combinations (except yellow) show the significant deviations (ANOVA, $p < 0.05$) from the null-hypothesis that the ARG has no influence on the mutant distribution across at least two replicates. Combinations highlighted in red illustrate the cases where the ARG shields the selection of the mutants that would otherwise be selected. Blue highlights situations in which an ARG that does not confer resistance to the antibiotic tested reduces the selection of the mutant; suggesting a negative interaction between the mutant and the ARG. Cases highlighted in yellow demonstrate the selection of the resistant mutant despite the presence of a resistance gene conferring resistance to the antibiotic tested, suggesting a dominant effect of the mutant over the resistance gene. ARG and mutant labels are coloured according to the expected main resistance conferred. AMK (amikacin), AZM (azithromycin), AZMCHL (azithromycin and chloramphenicol), CIP (ciprofloxacin), CST (colistin), DOX_2

(doxycycline – low resistance), DOX_6 (doxycycline – high resistance), FEP (cefepime), GEN (gentamycin), LB (Medium without antibiotics), TET (tetracycline), TMP (trimethoprim).

And main text from line 127:

*“In absence of antibiotic selection we observe no changes in the mutant abundance profile with introduction of ARG (ANOVA, $P > 0.05$), **highlighting that the resistance genes confer no or only minor fitness costs to the bacteria.** With antibiotic selection we observe 10 of 88 cases in which the mutant abundance profile is significantly (ANOVA, $p\text{-value} < 0.05$, Bonferroni corrected) changed by the introduction of a particular ARG. **It should be noted, that the majority of ARG-mutant combinations do not affect each other even under selecting condition. Under these conditions, mutants with the highest fitness in presence of the antibiotic and a MIC above the drug exposure were dominating. In most cases, those mutants were adapted to the antibiotic they were exposed to. However, in a few instances we also observed the selection of a mutant that was adapted to another drug but shows cross-resistance to the drug tested. For example in Cefepime mutants evolved to Doxycycline were preferentially selected even though the Cefepime evolved lineage had a higher MIC in Cefepime and comparable fitness in LB. This could be explained by changes in fitness in the presence of the antibiotic or by population dynamics between different mutants in the mutant pool. Unexpected selection has been previously reported for Cefepime (Jahn et al. 2018), highlighting that population dynamics can impact selection patterns. From the 10 observed interactions, the most common change (eight out of 10 cases) was the selection of a mutant distribution similar to the non-selective (LB) condition. This could result from the ARG shielding the effect of the antibiotic on the mutant pool when it confers resistance to the given antibiotic. One case of mutant dominance was also observed for the *qnrS1* gene present in the *gyrA* ciprofloxacin resistant background. **In this case the antibiotic resistance gene did not shield the selection of the mutant.** Finally, a case of strong negative epistasis was observed for the *tetA* gene in the amikacin evolved background (Figure 2).”***

In Fig.3, I think that there is a mistake: the FloR and Tem-219 cases should be exchanged, I guess (that is (Fig3A) and (Fig3B) are exchanged). But again, it is quite difficult to understand the meaning of this Fig.. This figure is almost not discussed. For example, when the antibiotic concentration is zero, one should expect that the barcoded mutants left should be the ones with higher fitness as shown in “Supplementary Figure 2. Unselected mutant fitness”, right? This should be explained. In Fig. 3B (should be FloR, I guess, as explained above), when the concentration of chloramphenicol is 2 ug/ml is this what we can see in Fig2? In that case, are is green zone the CIP-evolved and the grey zone the LB-evolved, and is this the reason why these squares are darker in Fig.2? All these things should be clearly explained.

We are grateful that the reviewer found that Fig3A and B should be interchanged and we provide an updated figure 3.

We agree with the reviewer that the fittest mutants are selected when the concentration is 0. We added this information along with a more detailed explanation of the figure to the text from line 159:

*“To investigate this phenomenon further, we determined mutant selection patterns in the presence or absence of the ARGs $bla_{TEM-219}$, $floR$ and $qnrS1$ at concentrations ranging from sub-MIC levels to 256 times the WT MIC of cefepime, chloramphenicol and ciprofloxacin, respectively. **When no antibiotic was present, the fittest mutants (LB, CIP and TMP evolved) were selected, as highlighted also in Figure 2.** From the selection patterns observed at the different drug concentrations, it is clear that mutants were selected for at very low (sub-MIC) drug concentrations in the absence of a protective ARG (Figure 3). A pattern of concentration-dependent mutant selection was observed, which corresponded well with the relative fitness cost of each mutant in the absence of antibiotics, as well as the respective antibiotic resistance profile (Supplementary Figure 2 and Supplementary Table 3). **Mostly, the mutants were selected at a respective concentration where they had the highest fitness in LB combined with a MIC above the drug exposure level. For example, above the MIC of chloramphenicol the mutants evolved to Doxycycline are dominating (Figure 3B) as observed previously (Figure 2).***

*For mutants carrying the $bla_{TEM-219}$ or $floR$ resistance genes, the presence of the ARG substantially enhanced the resistance of all of the mutants of the pool, resulting in the selection of mutants based on their fitness rather than on their resistance level to the drug over a wide concentration range. This is consistent with the ARGs shielding **or dominating** the impact of the antibiotic within a specific concentration **range corresponding to the resistance range of the ARG** (Supplementary Figure 3 A and B). **At drug concentrations just above the MIC, still the fittest mutants (LB, CIP, TMP evolved) were selected as also noted in Figure 2.** However, at high drug concentrations, when the antibiotic exposure approached the resistance level conferred by the ARG, the resistance gene no longer shielded the mutants, and differential selection of mutants resistant to the specific antibiotic was observed (Figure 3 A and B), **suggesting an additive effect of mutants and ARGs at these***

drug concentrations. Mutant selection resembled the selection without ARGs present at lower drug concentrations.

*Interestingly, the qnrS1 ciprofloxacin resistance gene confers lower resistance to ciprofloxacin than the low-cost gyrA (ciprofloxacin-adapted) mutation and does not provide substantial protective benefits to sensitive hosts (Figure 3 C). Because the gyrA mutant is highly resistant and its fitness cost is low, the resistance gene adds only a minimal amount of resistance to the mutant pool. Therefore, only a marginal shift in mutant selection is observed in the presence of qnrS1, resulting in a small additive effect dominated by the resistant mutant (Supplementary Figure 3). **Explaining, why no difference in selection was noted in Figure 2 for the qnrS1 gene.***

In addition, we did a series of pairwise competition experiments to validate and better explain the results shown in figure 2 and 3. Line 195:

“To validate selected experiments performed using our pooled competition approach, we additionally performed traditional pairwise competitions as previously described (Supplementary Figure 4 and Supplementary Table 4)20 (Silva et al. 2011). From the results, we observe similar dynamics to those obtained in our pooled competitions e.g. strong negative epistasis of tetA in the amikacin adapted mutant competed in amikacin and similarly for the antibiotic conditions where mutants were combined with protective ARGs. We also confirmed the concentration dependent selection of the dox6, azmchl and fep mutants observed in fig. 3A, where the dox6 mutant wins the dox-feb pairwise competition only at lower cefepime concentrations and the azmchl mutant is slightly fitter than both dox6 and fep mutants at higher (4 µg/ml) cefepime concentrations (supplementary fig 4B).

In addition, the pairwise competitions agree with our pooled fitness data where the ARG-mutant combinations did not show significant ($P > 0.05$) epistatic interactions in the absence of antibiotics (supplementary fig 4A and supplementary table 4).”

Assuming that my interpretation of Fig. 2 is correct and that Fig3A and Fig3B are exchanged, I now have some scientific concerns.

1) In Figure 2, why did you choose the concentration 2xMic? Why this value? What did you observe with other concentrations.

We thank the reviewer for questioning the reasoning behind our experimental design. We chose a concentration just above the MIC to maximize our chances of seeing both positive interactions as well as negative interactions without extinction of potential strong negative epistasis cases. In addition, many of the mutants only confer resistance at low levels and their selection dynamics would be lost at higher drug concentrations. Antibiotic concentrations close to, and even below the MIC, have been speculated to have a profound impact on the evolution of antibiotic resistance in the environment and the clinic (see e.g. PMID: 29686259 and others from the Andersson lab). We have elaborated on this in the text from line 89:

*“Initially, all ARG-mutant pools were subjected to antibiotics at concentrations just above the minimal inhibitory concentration (MIC) of the WT strain carrying the empty vector backbone, and at the highest drug concentration with growth were selected for further characterization for each antibiotic tested. **These concentrations were chosen to monitor all possible interactions while minimizing the chances of extinction of mutants due the low level of resistance generally conferred by mutations or in case of further reductions caused by strong negative epistasis.***

2) In Figure 2, why didn't the authors discount what happened with the LB situation?

We did several versions of the figure, including normalisation by subtracting the LB situation, however, we decided that showing the actual data and letting the reader interpret it would be easier to interpret and would give a better transparency of our findings. For example, subtracting the LB situation would highlight the most common selection pattern of the resistant mutants e.g. the negative selection of the LB, cip and tmp evolved in each antibiotic, while the shielding dynamics of the ARGs, or the pattern observed for tetA in the amk mutant, would be diminished and thus hard to observe.

3) In some places, MIC is referred and, in others, ug/ml is mentioned.

As the MIC is more intuitive and an easier point of reference when referring across drugs, we used the MIC in our general descriptions in the text. Yet, to display our data as precise as possible, when describing the selective condition for single drugs, we used ug/ml for the figures. To make it easier for the reader to relate the MIC to the actual concentrations, we added a line relating the concentrations to the MIC in Figure 3:

4) In Fig 4A, I think the legend is not very clear, given what it is in the Y-axis. In the legend, one can read “Doubling-time of the Amk4...”, but then, the Y-axis of Fig 4A is “Fold tetA induced growth reduction”. Is it “doubling time” or the factor by which the doubling time increased?

We appreciate that the reviewer mentions this important discrepancy and we corrected the figure legend accordingly:

“(A) **Fold tetA induced growth reduction of the Amk4 mutant grown in media in the presence (STR16) or absence (LB) of a sub-MIC concentration of streptomycin (16 µg/ml).**”

5) In Fig 4C, one can see the relative growth reduction of the tetA-carrying nuoF mutant. But is this really what the authors meant? In the figure, we see that at high pH, bars are lower. Does this mean that reduction is lower (that is, increase in the tetA-conferred sensitivity with increasing pH)?

We thank the reviewer for highlighting this imprecision. We updated the figure and figure legend and changed the Y-axis and describing text from relative growth reduction to relative growth.

6) I find it interesting that there is an overall positive effect when mutations and conjugative plasmids are together (Silva et al, ref 20) but that now the authors observe negative epistasis between mutations and ARGs. Does this imply that the backbones of plasmids are, somehow, helping ARGs to be maintained in bacterial cells (conferring lower fitness cost)?

We agree with the reviewer that this is a very interesting hypothesis. From our work on plasmid cost and survival we do believe that the reviewer is likely to be correct in that the backbones will help for the maintenance of resistance genes. From literature on plasmid evolution, resistance genes are rarely the locus of the fitness increases observed in plasmid adaptations, but more often mutations take place elsewhere in the plasmid backbone or host chromosome; supporting the reviewer’s line of thought. (PMID: 27501945, 22564249, 25302567). However, plasmids are very different in their genetic contents and we would also assume that these effects are highly depended on the exact backbone, host genetics (including co-residing plasmids) and environmental factors; thus generalizing from such studies is not a trivial task.

7) Title: I think that the title of this manuscript is misleading. Given that, when an ARG arrives into a cell, it is often travelling in a conjugative plasmid. In this case, there is no limitation of co-selection of vertically and horizontally acquired antibiotic resistance factors: ARGs do not arrive alone.

Good point. We changed the title of the manuscript to:

“Dominant resistance and negative epistasis can limit the co-selection of de novo resistance mutations and antibiotic resistance genes”

Reviewer #2

The manuscript studies the interaction between chromosomal mutations and horizontally acquired resistant determinants in the evolution of antibiotic resistance. The authors combined 11 different chromosomally barcoded E. coli mutants adapted to different antibiotics with 11 different antibiotic resistance genes (ARGs) present on a medium copy plasmid and measured the fitness of the mutant-ARG combinations in the presence of 8 different antibiotics. At relatively low antibiotic concentrations (twice the MIC of the wild type carrying the empty vector) they identified significant interactions for approx. 12% (10 out of 88) of the mutant-ARG combinations and in 80% of these cases ARGs dominated the resistance phenotype of the mutant, shielding the selective effect of the antibiotic on the mutant pool.

The authors reach the conclusion that at low antibiotic concentrations the resistance determinant with the highest resistance level will dominate and shield the selection dynamics of the other resistant determinant, while at high antibiotic concentrations both determinants will be selected for. In one case they identified negative epistasis between two unrelated resistance mechanisms, the tetA efflux pump and loss of function nuo mutations involved in aminoglycoside resistance. By the mining of 13500 E. coli genomes they found that tetA was never observed with any loss of function mutations in the nuo pathway, supporting the negative epistasis.

Overall, I appreciate the systematic approach that integrates vertically and horizontally acquired resistance determinants to study resistance evolution and I find the results to be of potential broad interest.

We thank the reviewer for her/his interest in our study and for the time spent reviewing the manuscript.

However, the clinical relevance of the work is questionable due to the low antibiotic dosage employed. Also, I wish to highlight that a paper with similar goals have recently been published (Silva et al, Plos Genetics, 2011). Given major differences in the methods used to evaluate epistasis between the two studies, it is difficult to reconcile why different conclusions were reached.

We thank the reviewer for the critical thoughts regarding our experimental setup. However, we do believe that there is plenty of literature supporting the relevance of low (including sub-MIC) concentrations of antibiotics in the evolution of clinically relevant antibiotic resistance in clinical and environmental settings (e.g. from the Andersson lab - Pubmed Ids: 24861036, 21811410, 29686259). In addition, the highest concentrations chosen for sequencing were close to the clinical breakoins of the antibiotics. To be more transparent in

this point, we added the clinical breakpoint to table S6.

As highlighted several times in the manuscript, we are well aware of the study by Silva et al, and agree that their findings are very interesting. However, the authors focus on a few mutants and their interactions with complex plasmid backbones (where it is hard to isolate effects and generalize to specific genotypes), where our goal was to more specifically decipher the direct interaction of resistance mutations with resistance genes. In the light of our findings we believe that the differences between our study and the findings by Silva et al. 2011 are mainly attributed to other (still unknown) factors of the large (>60kb) plasmid backbones investigated by Silva et al, and not direct mutant-ARG interactions, as we have highlighted in the discussion of the study by Silva et al. in our manuscript Line 329:

"...our study focused specifically on ARG-mutant interactions and suggests that such effects are unlikely to stem from ARG-mutant interactions, but rather from the remaining portion of the large plasmid backbones. This notion is supported by previous studies of plasmid-host evolution that describe interactions between the host chromosome and several plasmid components not related to antibiotic resistance^{6,26,27}."

Major comments:

Pooled competition assays are appropriate as a screening method to have a first and rude approximation of bacterial fitness. However, for the identification of exact epistatic interactions, pairwise competition assays (i.e. wild-type versus antibiotic resistant bacteria) are needed to reach precise relative fitness values both in the presence and absence of antibiotic. Based on these precise fitness values, the nature of interactions (negative, positive or sign epistasis) can be determined and deviation from the null model (additive interaction) should be evaluated statistically. I find it critical to verify the nature of epistatic interactions in a rigorous manner, see e.g. Silva et al, Plos Genetics, 2011. I believe this should be done for several mutant-plasmid pairs.

Although we do believe that one should be careful comparing our results directly to the data obtained in the study by Silva et al. (due to the complexity of natural plasmids as discussed above), we agree with the reviewer that validation of selected observations through direct competition would be beneficial to perform.

However, we would like to note that the interactions between bacteria in nature rarely happens between two species/strains alone, but are much more likely to occur in community settings more closely represented by a pooled competition assay.

To validate our results and allow comparison with previous studies like Silva et al. 2011, we performed a series of 28 individual competition experiments encompassing ARGs, mutants and their combinations competed with and without antibiotics.

Due to more specific questions from the other reviewers and different interaction patterns observed, we selected the amk, cip, feb, dox and azmchl mutants along with the tetA, tem219 and qnrS1 ARGs for the competition experiments.

The experiments and calculations were performed as described in Silva et al 2011 and the results are depicted in the supplementary material (supplementary fig 4).

We have added to the manuscript (Line 205):

" From the results we observe similar dynamics to those obtained in our pooled competitions

e.g. strong negative epistasis of *tetA* in the amikacin adapted mutant competed in amikacin as well as for the antibiotic conditions where mutants are combined with protective ARGs. We also observed the concentration dependent selection of the *dox6*, *azmchl* and *fep* mutants observed in fig. 3A, where the *dox* mutant wins the *dox*-*fep* pairwise competition only at lower cefepime concentrations and the *azmchl* is slightly fitter than both *dox6* and *fep* at higher (4ug/ml) cefepime concentrations (supplementary fig 4B and C). In addition, the pairwise competitions agree with our pooled fitness data where the ARG-mutant combinations did not show significant ($P > 0.05$) epistatic interactions in the absence of antibiotics (supplementary fig 4A and supplementary table 4). “

We have added graphs displaying the data to the supplement:

Supplementary Figure 4.

Fitness measures resulting from pairwise competition experiments. Each strain was competed against the WT MG1655 for 24h in either LB (A) or LB + antibiotics (B) and fitness is shown relative to this strain. Additionally, direct competitions of mutants *dox6* against *fep* (*Dox6/Fep*) and *fep* against *azmchl* (*Fep/AzmChl*) were performed for the cefepime experiments. For cefepime, two concentrations of 0.06 ug/ml (turquoise) and 4 ug/ml (purple) were used. For ciprofloxacin and amikacin, the same concentrations as used for the pooled competitions displayed in fig. 2 (0.0075 ug/ml and 16 ug/ml) were used. Error-bars show the standard deviation of three biological replicates. The stippled horizontal line represents the point where the competing strains are equally fit.

As well as a table with relevant epistasis and p-values:

Supplementary Table 4.

Epistasis values of individual competitions. Epistasis (calculated as: “Fitness of mutant with ARG – fitness of ARG in WT * fitness of mutant”) and propagated errors were calculated according to Silva et al. 2011 (reference 20).

Strain	Epistasis	p-value	Condition
Amk(tetA)	0.030	0.636	LB

Cip(qnrS1)	-0.039	0.341	LB
AzmChl(Tem219)	0.002	0.506	LB
Fep(Tem219)	-0.027	0.284	LB
Amk(TetA)	-0.304	0.044	Amk
Cip(QnrS1)	-1.140	0.005	Cip
AzmChl(Tem219)	-0.442	0.049	Fep (0.06 ug/ml)
Fep(Tem219)	-0.490	0.029	Fep (0.06 ug/ml)

And a brief description of the methodology in the methods section (line 424):

Pairwise competition experiments

Head to head competition experiments were performed as described in Silva et al.²⁰ Briefly: Strains were grown O/N in LB and diluted 10^{-4} before mixing in 10 ml LB. All competitions were done in three biological replicates. After initial plating to quantify starting numbers, the competitions were incubated at 37 °C for 24 h with shaking. The reference strains used here was *E. coli* K12 MG1655 Δ lacZ (created by introducing a nonsense mutation in the lacZ gene) and CFU of the competing strain and reference was enumerated as blue and white colonies, respectively, on LB plates containing X-Gal (Sigma). The lacZ mutant did not show a significant fitness cost ($P = 0.57$) when competed against its ancestor. Four competitions were conducted with direct competition of mutants dox6 against fep and fep against azmchl in cefepime and these were distinguished based on differences in their resistance profiles.

The conclusions drawn from a competition assay performed at twice the wild type MIC antibiotic concentration. Is this relevant? I think it is important to repeat the high-throughput pooled competition assay closer at much higher dosages. Does epistasis depend on the dosage used? If not, this should be justified in systematic manner. I understood correctly, mutant pools were sequenced at the highest antibiotic concentration too, so this can be easily verified. In addition, pairwise epistatic interactions should be determined as well at these higher antibiotic concentrations to reach clinical relevance with the results. Related to clinical relevance, I missed the information whether the resistance level of the used antibiotic adapted lines is clinically relevant (above the clinical breakpoint) or not. Same question applies to the ARGs.

We thank the reviewer for questioning the reasoning behind our experimental design. We chose a concentration just above the MIC to maximize our chances of seeing both positive interactions as well as negative interactions without extinction of potential strong negative epistasis cases. In addition, many of the mutants only confer resistance at low levels and

their selection dynamics are therefore more informative at lower drug concentrations that we do believe to be of clinical relevance (see comment on sub-MIC selection above). We have elaborated on this in the text from line 91:

*“Initially, all ARG-mutant pools were subjected to antibiotics at concentrations just above the minimal inhibitory concentration (MIC) of the WT strain carrying the empty vector backbone, **which was generally close to the clinical breakpoints for the selected drugs (Supplementary table 6)**, and the highest drug concentration with growth were selected for further characterization for each antibiotic tested. **These concentrations were chosen to monitor all possible interactions while minimizing the chance of extinction of mutants due the low level of resistance generally conferred by mutations, or in case of further reductions caused by strong negative epistasis.**”*

Higher concentrations were sequenced that cover the higher antibiotic concentrations of the clinically relevant spectrum of concentrations and even above (the highest concentration with observable growth). In most cases however, the conclusions did change at these higher concentrations, and fig. 3 illustrates the cases where the most interesting dynamics were observed. For these concentration dependent cases, we sequenced the mutant pools at several concentrations as shown in Figure 3, where dose dependency of the interactions is illustrated and concluded on in the discussion line 341:

“The coexistence of ARGs and chromosomal mutations that confer resistance towards the same condition are more likely to be selected for if they interact in a favorable manner to increase overall resistance or fitness. In such cases, we observed an additive effect of ARG-mutant combinations where the determinant with the highest resistance level would shield the selective dynamics of the other element at low selection levels whereas both elements would be selected for at high concentrations. “

In a similar manner we show that the epistasis of the tetA gene and nuo-mutants is dose (and pH) dependent in fig. 4.

To further support this we conducted pairwise competition experiments in low and high antibiotic concentrations between the dox6 against fep (Dox6/Feb) and fep against azmchl (Fep/AzmChl) mutants that showed interesting selection dynamics in fig 3 (See supplement fig. 4 B).

Finally, we also added fold clinical breakpoint to the supplementary table 6 that provide resistance information of the mutants and genes.

The authors claim in the legend of figure 2 that “the ARG shields the selection of the mutant that would otherwise be selected.” It is not clear what the initial expectation about the selection of mutants is, if there is any expectation. Do we expect those mutants to be selected i) that were already adapted to the given antibiotic (this happens in 5 out of the 8 cases), ii) that have the highest fitness in the absence of any antibiotic or iii) that show cross-resistance to the given antibiotic and have a high fitness (e.g. doxycycline mutants)? For example, what can be the possible explanation that cefepime adapted mutants are not selected in the presence of cefepime but doxycycline mutants are, when cefepime mutants have the highest cefepime MIC and the fitness of the two mutants is very similar in the absence of antibiotic selection. Similar question applies to tetracycline adapted lines.

We thank the reviewer for thoroughly going through our results and for requesting additional explanations that will clarify our findings. As the reviewer points out, selection is complex and can be broken down into components such as non-selective fitness and antibiotic resistance levels. We observe different cases, most of these being the first mentioned by the reviewer where the selection of the most highly resistant clones is favoured; however when a resistance gene is present, the more important selective component turns to the fitness (as measured in the absence of antibiotics).

Because many mutants are not specific to the drug in which they were selected (and resistance can arise from adaptation to antibiotic unrelated conditions, see PMID: 28893783), predicting their selection patterns across drugs is not straight forward, but nevertheless interesting. While going more deeply into the selection dynamics of mutants alone is beyond the scope of this study, we agree that selection of the dox mutant over the fep mutant is surprising. Therefore we included these mutants in our pairwise competition experiments described above.

While the phenomenon is not completely understood it deserves more thorough consideration in the text:

line 128:

“With antibiotic selection we observe 10 of 88 cases in which the mutant abundance profile is significantly (ANOVA, p -value<0.05, Bonferroni corrected) changed by the introduction of a particular ARG. It should be noted, that the majority of ARG-mutant combinations do not affect each other even under selecting condition. Under these conditions, mutants with the highest fitness in presence of the antibiotic and an MIC above the drug exposure were dominating. In most cases, those mutants were adapted to the antibiotic they were exposed to. However, in a few instances we also observed the selection of a mutant that was adapted to another drug but shows cross-resistance to the drug tested. For example in Cefepime mutants evolved to Doxycycline were preferentially selected even though the Cefepime evolved lineage had a higher MIC in Cefepime and comparable fitness in LB. This could be explained by changes in fitness in the presence of the antibiotic or by population dynamics between different mutants in the mutant pool. Unexpected selection has been previously reported for Cefepime (Jahn et al. 2018), highlighting that population dynamics can impact selection patterns.”

To have a clear picture, I would suggest the de novo construction of nuoH and nuoF single mutants starting from the wild type background and verify in the conventional way that indeed they show negative epistasis with tetA. Then test the sensitivity of these mutants against different aminoglycosides in the presence of TetA to clarify if the observed negative epistatic interaction applies generally to aminoglycosides or not.

We thank the reviewer for raising this point. The construction of nuo mutants can be very challenging as they confer only low resistance, wherefore it is hard to select for a phenotype. Ultimately, we believe that the nuo+fusA mutant vs. the reconstructed fusA mutant, along with the effect being observed in for different strains with distinct nuo mutations (nuoH and nuoF), is sufficient in showing that the effect is due to the mutation in nuo.

When the authors tested the effect of the growth medium pH on the tetA-conferred streptomycin sensitivity, they claim that tetA-conferred sensitivity increased with increasing pH. However, based on figure 4C, with the exception of the highest streptomycin concentration, no clear trend can be observed. I would suggest repeating the experiment using a broader pH range. It also remains an open question why these particular (5.8, 6.7, 7.9) pH values were chosen and it is not clear what data is visualized on the y axis of the figure. What does the red dotted line show? The growth of the mutant carrying an empty vector? If yes, then the relative growth of tetA-carrying nuoF mutant is shown on the y axis.

We thank the reviewer for the concerns regarding our pH-dependent growth experiment. We have corrected the label on the y-axis to reflect the actual measure "Relative growth". We aimed at pH values of 6, 7 and 8, however, the pH drifts slightly depending on timing and temperature conditions so we prefer to show the exact pH values measured just before the onset of the experiment. We did start to observe fairly strong growth defects in our E. coli strain when going above the selected pH range (which would also be considered unnatural for E. coli), and we would therefore not increase the pH any further.

We agree with the reviewer that we cannot claim a strong direct correlation between pH and the sensitivity to streptomycin, however we do believe that the increasing sensitivity with increasing streptomycin concentration at the highest pH, which is not observed at the lower pH levels, is a very strong indication of this.

We have moderated our claims on the pH dependence in the text (line 282):

"By measuring the growth reduction relative to growth in the tetA-free background at three streptomycin concentrations, we observed a pH dependent increase in the tetA-conferred sensitivity. "

Minor comments:

1) How exactly was the relative abundance of each mutant within each ARG-associated mutant pool calculated?

The abundance was calculated based on sequencing reads as stated in the Materials and Methods in line 396:

"The barcode frequencies for each condition were determined as the number of reads containing each barcode relative to the total number of barcoded reads."

2) It is not clear what NoVec stands for in supplementary figure 1, if relative abundances of ARG-mutant combinations are shown normalized to the fitness of mutants carrying the empty vector.

NoVec stands for vector-free control, which represents the mutant pool without any plasmid. We have clarified this in the figure legend.

3) In figure 3 Tem-219 and floR are interchanged. It would be informative to highlight the 2xMIC concentration in this figure.

We updated the figure and added the MIC concentration of the WT as a reference line to the figure.

4) Supplementary figure 4 is very difficult to interpret. Please provide more information in the figure legend, for example what does the red dotted line represent and what exactly is shown on the y axis?

We thank the reviewer for pointing this out. The figure legend has been updated:

“Sensitivity of the tetA carrying Amk4 mutant towards sub-MICs of gentamicin and amikacin (8 µg/ml), shown as the impact on growth. The observed sensitivity is only significant for amikacin (Wilcoxon rank sum test, $P < 0.05$). The red dotted line highlights equal fitness (no effect of tetA).”

5) What was the reason to incubate the cells without shaking in the competition experiment?

The setup of our competition experiment resembled conditions for MIC determination (11-step 2-fold dilutions of antibiotics in 96-well plates, incubated not shaking for 18 h at 37 C). We chose this setup, so that we could relate of information on the MICs of the different mutants better to the results of the competition experiment.

Reviewer #3

In general, the idea of looking for genetic interactions between resistance genes and chromosomal resistance mutations is quite interesting. The authors performed a screen for this based on previously generated experimental evolution, and then focused on a specific interaction between tetA resistance gene and nuo mutations involved in aminoglycoside tolerance. The computational analysis showing that these two mutations never co-occur despite expectation was quite interesting.

We thank the reviewer for her/his interest in our work.

My biggest issue with this paper is that the experiments in Figure 1 suggest that interactions between vertically and horizontally are not very common, but the title and abstract seem to imply the opposite. I am concerned the result of this is to oversell the generality of the results. The specific interaction explored, however, is quite interesting, and the paper would likely be far better off claiming that well-supported finding as a main result and not attempting to overgeneralize.

We thank the reviewer for the critical input. We agree that the findings might have been over generalized and we have now toned down and specified our language several places in the manuscript:

The new title reads:

“Dominant resistance and negative epistasis can limit the co-selection of de novo resistance mutations and antibiotic resistance genes”

And we updated the abstract I line 22:

“While the majority of interactions is neutral we identify significant interactions for 12 % of the mutant-ARG combinations.”

And toned down the concluding paragraph of the abstract line 30:

“Our study highlights important constraints that may allow better prediction and control of antibiotic resistance evolution.”

And in the overall conclusion line 371:

“In conclusion, while the coexistence of most interaction pairs was not constrained, our data demonstrates existence of several interactions between ARGs and chromosomal resistance mutations that may affect their dissemination.”

Specific Comments:

- Fig 2 was hard to understand, it took a few attempts to really get it. Also, showing log (fold change in frequency) for each strain might be easier to interpret than relative abundance. It might also be useful to have a metric for comparing abundance profiles.

We updated the figure and hope that it is more intuitive to understand now. We appreciate the reviewer's suggestions, however, we doubt that a log scale would be helpful in order to understand the frequencies better and we believe that the data is more transparent in its current form (PMID: 30013132). We did several versions of the figure, including normalisation by subtracting the LB situation, however, we decided that showing the actual data and letting the reader interpret it would be easier to interpret and would give a better transparency of our findings. For example, subtracting the LB situation would highlight the most common selection pattern of the resistant mutants e.g. the negative selection of the LB, cip and tmp evolved in each antibiotic, while the shielding dynamics of the ARGs, or the pattern observed for tetA in the amk mutant, would be diminished and thus hard to observe.

- Line 133: I'm afraid I do not know what it means to "dominate in an additive fashion" -- it's clear what it means when the effect of one mutation dominates another, which is what I believe the authors intend here. There is a small bit of explanation on lines 159-60, but I think it would help a broader audience understand the results if the authors took some care to be a bit more explicit and precise in their claims here.

We appreciate that the reviewer highlights this shortcoming in our explanations and we updated the manuscript to accommodate requested changes and additional explanations.

Line 153:

“ARGs tend to dominate resistance mutants within the resistance range of the ARG”

Line 176:

*“This is consistent with the ARGs shielding or dominating the impact of the antibiotic within a specific concentration range **corresponding to the resistance range of the ARG** (Supplementary Figure 3 A and B). **At drug concentrations just above the MIC, still the fittest mutants (LB, CIP, TMP evolved) were selected as also noted in Figure 2.** However, at high drug concentrations, when the antibiotic exposure approached the resistance level conferred by the ARG, the resistance gene no longer shielded the mutants, and differential selection of mutants resistant to the specific antibiotic was observed (Figure 3 A and B), **suggesting an additive effect of mutants and ARGs at these drug concentrations. Mutant selection resembled the selection without ARGs present at lower drug concentrations.**”*

- In Fig 3A, 3B: authors claim that the mutant pool profile below a certain amount of antibiotic does not depend on antibiotic concentration. While the claim is plausible, the authors interpolate between data points that are quite widely spaced (between 0 and 2, 2 and 16 in 3A, 0 and 12 in 3B). It would be useful to have data for the intermediate points, or to weaken the claim.

We thank the reviewer for pointing this out and we have weakened our claim in paragraph from line 173:

*“For mutants carrying the $bla_{TEM-219}$ or $floR$ resistance genes, the presence of the ARG substantially enhanced the resistance of all of the mutants of the pool, resulting in the selection of mutants **primarily** based on their fitness rather than on their resistance level to the drug **for the lower drug concentrations included here.**”*

- Line 201: Sentence starting with “To confirm the involvement” was a confusingly worded and it took some time to understand what strains were used in the competition experiment.

We thank the reviewer for this observation and we have changed our wording of the sentence from line 251:

*“**We further tested** the involvement of the nuo mutant in this sensitive phenotype, **by conducting a pair-wise competitive fitness experiment, where the tetA-carrying $nuoF$ and $fusA$ mutant was competed against a tetA-carrying mutant with the same $fusA$ background but with the $nuoF$ allele reverted to WT (Figure 4 B).**”*

- Line 205: “Additionally, the presence of tetracycline did not have a material effect on the selection patterns”: there appears to be no data to support this claim.

We believe that Figure 4B (the y-axis) shows that there is no significant invariance between different tetracycline concentrations, serving as data to back up our claim.

- 2D competitive fitness assay (Figure 4B and 397-404): While the results are strong, to establish the effect beyond a doubt it would be good to repeat the experiment with dye-swapped strains. I do not

think this is strictly necessary, however, as the likelihood of there being a differential effect on drug resistance between gfp and rfp is minimal.

We thank the reviewer for the suggestion, however, we believe that the strong effect observed in the competition experiment along with the data generated through our multiplexed competitions, pairwise competitions, and growth rate data involving the nuo mutants carrying tetA very strongly supports the validity of the results. In addition we conducted additional pairwise competitions (Supplementary figure 4) to further support the findings of fig. 4 B. Furthermore, the use of GFP and RFP tagged strains for fitness comparisons is widely used in the literature and GFP and RFP do not seem to cause strong differences in selection outcomes (see: PMIDs 26937640, 29632354, 30728359).

- Lines 230-232 ("These results suggest... limit their co-selection in natural E. coli isolates"): That the lack of co-occurrence is due to aminoglycoside selection is a bit of a leap and is not supported. As many things depend on PMF, this pattern could be due to any number of selective pressures.

The paragraph was meant as a general concluding statement of the whole section, including the experimental results, and not just the bioinformatic analysis. To clarify this to the reader we changed the start of the sentence from line 297:

"Taken together, the experimental and bioinformatic results suggest that the presence of tetA alters the resistance level of aminoglycoside resistance mutants, carrying mutations in the nuo operon, in a PMF-dependent manner and that such interaction may limit their co-selection in natural E. coli isolates."

- Line 343: Why were these cultures grown without shaking? This is pretty nonstandard, as it allows within-well differentiation due to nutrient (and oxygen) gradients and wall-bound growth. While I don't expect this to bias the results too much, it may make them harder to reproduce.

We thank the reviewer for posing a relevant question. The setup of our competition experiment resembled the standard conditions for MIC determination (11-step 2-fold dilutions of antibiotics in 96-well plates, incubated without shaking for 18 h at 37 C). We chose this setup, so that we could relate information on the MICs of the different mutants better to the results of the competition experiment. We also doubt that the reproducibility is severely impacted by the setup as our replicates displayed little variation in barcode frequencies.

To clarify this we have added to the results section of the manuscript line 402:

"Each mutant pool was subjected to a two-fold concentration gradient ranging over 11 different concentrations of eight different antibiotics, resulting in more than 3000 competition experiments in at least two replicates. This approach was used to mimic the methodology by which the individual MIC values were measured."

Reviewers' comments:

Reviewer #1 (Remarks to the Author):

I'm satisfied with this second version of the manuscript, as well as with the answers given by authors.

Minor revision:

- Lines 201-204 - The sentence "The fact that most ARGs confer high..." is too long and was confused about its meaning.

- Suggestion: all data should be made available as an excel file (.xls) or equivalent (e.g., data used to generate all Figs in the main text as well as to generate Suppl. Figs 4 and 5).

Reviewer #2 (Remarks to the Author):

I appreciate the authors' effort to validate their results by performing pairwise competition experiments. However, for many reasons, I found it quite difficult to compare and integrate these new results with the results coming from the pooled competition experiments and the results from Silva et al, 2011.

First, the way new data is presented makes the interpretation difficult. Instead of showing separately fitness data for different strains in Supplementary Figure 4 and epistasis values in Supplementary Table 4, it would be more straightforward to show the epistasis results in a 5x3 matrix, as for the validation experiments, the authors chose 5 mutants and 3 ARGs.

Second, in the presence of antibiotic all interactions were negative epistasis, while in the absence of antibiotic, no significant epistasis was found. How can these results be related to those of Silva et al, where positive epistasis dominated?

My other concern is related to the integration of the new data. In the case of AMK(TetA) mutant-ARG combination the situation is straightforward as the result coming from the pairwise competition (even if with a p-value very close to 0.05) is in accordance with the pooled competition results, showing negative epistasis. However, in the case of the other 3 combinations, comparison is not obvious. For example, taking the AzmCh(Tem219) case. Based on the pooled competition results, the authors claim that the ARG dominates mutation, while based on the pairwise competition they found negative epistasis between mutant and ARG. It needs an explanation. The same applies to Cip(QnrS1) and Fep(Tem219).

Minor issues:

What does Dox6/Fep abbreviation stand for? In the figure legend, it is stated that "direct competitions of mutants dox6 against fep (Dox6/Fep) and fep against azmchl (Fep/AzmChl) were performed for the cefepime experiments". How should the reader interpret the fitness value in these cases? Relative to the fitness of which strain? Please clarify.

Which panels correspond to Supplementary figure 4C mentioned in the main text?

The authors do not use consistently the Fep abbreviation (sometimes they use Feb) for cefepime mutants.

Reviewers' comments:

Reviewer #1 (Remarks to the Author):

I'm satisfied with this second version of the manuscript, as well as with the answers given by authors.

Minor revision:

- Lines 201-204 - The sentence "The fact that most ARGs confer high..." is too long and was confused about its meaning.

We agree with the reviewer and have shortened and altered the sentence to be more clear; line 21-204:

"The fact that most ARGs confer high levels of resistance means that resistance mutations may often be redundant when ARGs are present."

- Suggestion: all data should be made available as an excel file (.xls) or equivalent (e.g., data used to generate all Figs in the main text as well as to generate Suppl. Figs 4 and 5).

We have compiled an excel file with all the relevant source data and uploaded with the submission.

Reviewer #2 (Remarks to the Author):

I appreciate the authors' effort to validate their results by performing pairwise competition experiments. However, for many reasons, I found it quite difficult to compare and integrate these new results with the results coming from the pooled competition experiments and the results from Silva et al, 2011.

We thank the reviewer for the suggestions on how to improve the clarity of our new results below. We hope that our corrections are satisfactory.

First, the way new data is presented makes the interpretation difficult. Instead of showing separately fitness data for different strains in Supplementary Figure 4 and epistasis values in Supplementary Table 4, it would be more straightforward to show the epistasis results in a 5x3 matrix, as for the validation experiments, the authors chose 5 mutants and 3 ARGs.

We thank the reviewer for the suggestion. The matrix (no AB selection) would look like:

No AB	tetA	qnrS1	tem219
Amk	0.030		
Cip		-0.039	

AzmChl			0.002
Fep			-0.027

And another would be needed for the antibiotic conditions. These have been included in the source data file.

However, given the amount of empty space in such a matrix, and the additional table needed to relate the epistasis values to their p-values and exact selective conditions, we would prefer to keep our current table layout in the supplement document.

Second, in the presence of antibiotic all interactions were negative epistasis, while in the absence of antibiotic, no significant epistasis was found. How can these results be related to those of Silva et al, where positive epistasis dominated?

Unlike Silva et al, we conducted our measurements directly on isolated ARGs in mutant backgrounds and the strong negative epistasis is observed only in the presence of antibiotics (which was not tested by Silva et al). As mentioned in the discussion, we believe that our results suggest that the positive epistasis observed by Silva et al. stems from the complex large (>60kb) plasmid backbones rather than direct interactions between ARGs and mutants.

Discussion line 374: “While a previous study²⁰ reported strong epistatic effects on fitness for combinations of conjugative plasmids with *gyrA*, *rpoB* and *rpsL* mutants, our study focused specifically on ARG-mutant interactions and suggests that such effects are unlikely to stem from ARG-mutant interactions, but rather from the remaining portion of the large plasmid backbones. This notion is supported by previous studies of plasmid-host evolution that describe interactions between the host chromosome and several plasmid components not related to antibiotic resistance^{6,26,27}. Contrary to the study by Silva et al.²⁰ we specifically assessed the ARG-mutant interaction and did not find any significant effect on fitness in the absence of antibiotics. While it is possible that our 24 h competition assay could not detect minor effects on combined ARG-mutant fitness, we were able to detect the subtle fitness differences between low-cost mutants e.g., in those with mutations in *folA* and *gyrA*, compared to the susceptible ancestor (**Supplementary Figure 2**), suggesting that that potential effects missed by our assay were minor.”

My other concern is related to the integration of the new data. In the case of AMK(TetA) mutant-ARG combination the situation is straightforward as the result coming from the pairwise competition (even if with a p-value very close to 0.05) is in accordance with the pooled competition results, showing negative epistasis. However, in the case of the other 3 combinations, comparison is not obvious. For example, taking the AzmCh(Tem219) case. Based on the pooled competition results, the authors claim that the ARG dominates mutation, while based on the pairwise competition they found negative epistasis between mutant and ARG. It needs an explanation. The same applies to Cip(QnrS1) and Fep(Tem219).

As exemplified in figure 3, mutations tend not to add any additional resistance when an ARG is present (negative epistasis), unless the resistance capacity (generally far above the clinical breakpoint) of the ARG is exceeded. The pairwise competitions mentioned by the reviewer were done at antibiotic concentrations used in fig. 2 (indicated by the dashed

line in fig. 3), where no additive effect of mutations is observed, and are therefore perfectly comparable and aligned with these results.

This is mentioned several places in the text:

Line 24: “We found that the ability of most ARGs to confer high level resistance at a low fitness cost shields the selective dynamics of mutants at low drug concentrations leading to the selection for high fitness mutants regardless of their resistance level.”

And we have clarified it further in the text:

Line 186: “This is consistent with the ARGs shielding, or dominating, the impact of the antibiotic within a specific concentration range corresponding to the resistance range of the ARG (**Supplementary Figure 3 A and B**); after which additive effects of ARGs and mutants are observed.”

And in the section on pairwise competitions Line 207:

“Similarly, we could confirm the strong negative epistasis observed between mutants and ARGs for the antibiotic conditions used in Fig. 2”

And amended the discussion to accommodate our interpretation of the results. Line 383:

“In such cases, we observed an additive effect of ARG-mutant combinations only when antibiotic concentrations were high (generally above the clinical breakpoint). The negative epistatic interaction between ARGs and mutants at lower drug concentrations observed here, may therefore encourage reversion to susceptible genotypes when ARGs are present in environments exposed to drug concentrations within their resistance capacity where most mutational resistance phenotypes do not add to survival.”

Minor issues:

What does Dox6/Fep abbreviation stand for? In the figure legend, it is stated that “direct competitions of mutants dox6 against fep (Dox6/Fep) and fep against azmchl (Fep/AzmChl) were performed for the cefepime experiments”. How should the reader interpret the fitness value in these cases? Relative to the fitness of which strain? Please clarify.

Generally, the fitness measures are given in relation to the WT strain and e.g. “Amk” could also be written Amk/wt. However the “wt” is omitted as this is the standard reference. When the wt is not used as a reference the direct competition is conducted between two mutants, and should be interpreted as a fraction of change in cfu counts resulting from the competition. I.e. the Dox6/Fep expresses how the numerator (Dox6 mutant) is changing during competition in comparison to the denominator (Fep mutant).

This has been clarified in the figure legend of supplementary figure 4:

“Additionally, direct competitions of mutants *dox6* against *fep* (*Dox6/Feb*) and *fep* against *azmchl* (*Fep/AzmChl*) were performed for the cefepime experiments. The fitness is expressed as the fraction is written.”

Which panels correspond to Supplementary figure 4C mentioned in the main text?

We are sorry about the confusion, but we cannot find any mention of Supplementary figure 4C in the text?

The authors do not use consistently the Fep abbreviation (sometimes they use Feb) for cefepime mutants.

Thanks for noticing. This has been corrected to the consistent use of “Fep”.